

# Reviews and syntheses: Anthropogenic perturbations to carbon fluxes in Asian river systems: Concepts, emerging trends, and research challenges

Ji-Hyung Park[1], Omme K. Nayna[1], Most S. Begum[1], Eliyan Chea[2], Jens Hartmann[3], Richard G. Keil[4],
Sanjeev Kumar[5], Xixi Lu[6], Lishan Ran[7], Jeffrey E. Richey[3], Vedula V.S.S. Sarma[8], Shafi Tareq[9], Do
Thi Xuan[10], Ruihong Yu[11]

[1]Department of Environmental Science and Engineering, Ewha Womans University, Seoul, 03760, Republic of Korea
[2]Department of Environmental Science, Royal University of Phnom Penh, Phnom Penh, Cambodia
[3]Institute for Geology, Universität Hamburg, Hamburg, Germany
[4]School of Oceanography, University of Washington, Seattle, 98112, USA
[5]Geosciences Division, Physical Research Laboratory, Ahmedabad, 380009, India
[6]Department of Geography, National University of Singapore, Singapore
[7]Department of Geography, University of Hong Kong, Pokfulam Road, Hong Kong
[8]National Institute of Oceanography, Council of Scientific and Industrial Research, Visakhapatnam, India
[9]Department of Environmental Sciences, Jahangirnagar University, Dhaka, 1342, Bangladesh
[10]Department of Soil Science, Cantho University, Cantho, Vietnam
[11]College of Environment and Resources, University of Inner Mongolia, Hohhot, China

*Correspondence to*: Ji-Hyung Park (jhp@ewha.ac.kr)

**Abstract.** Human activities are drastically altering water and material flows in river systems across Asia. These anthropogenic perturbations have rarely been linked to the carbon (C) fluxes of Asian rivers that may account for up to 40–50% of the global fluxes. The primary object of this review was to provide a conceptual framework for assessing human impacts on Asian river C fluxes, along with a latest update on anthropogenic alterations of riverine C fluxes, focusing on the impacts of water pollution and river impoundments on $CO_2$ outgassing from the rivers draining South, Southeast, and East Asian regions that account for the largest fraction of river discharge and C exports from Asia and Oceania. Recent booms in dam construction across Asia have created a host of environmental problems; yet only a small number of studies have explicitly investigated altered rates of greenhouse gas (GHG) emissions and organic C transport. There have been contrasting reports on impoundment effects: decreases in GHG emissions in the reservoirs exhibiting enhanced primary production vs. increased emissions from the flooded vegetation and soils in the early years following dam construction or from the impounded river reaches and downstream estuaries during the monsoon period. These contrasting results suggest that the rates of metabolic processes in the impounded and downstream reaches can greatly vary longitudinally over time, as a combined result of diel shifts in the balance between autotrophy and heterotrophy, seasonal fluctuations between the dry and monsoon periods, and a long-term change from a leaky post-construction phase to a gradual C sink. Rapid pace of urbanization across southern and eastern Asian regions has dramatically increased municipal water withdrawal, generating annually 120.2 km$^3$ of wastewater in 24 countries, which comprises 38.6% of the global municipal wastewater production



(311.6 km$^3$). Although the municipal wastewater constitutes only 0.9% of the renewable surface water, it can disproportionately affect the receiving river water, particularly downstream of rapidly expanding metropolitan areas, including eutrophication, increases in the amount and lability of organic C, and pulse emissions of $CO_2$ and other greenhouse gases (GHGs). As reviewed for three representative rivers (the Ganges, Mekong, and Yellow River), the lower reaches of these rivers and their polluted tributaries tend to exhibit higher levels of organic C and the partial pressure of $CO_2$ ($pCO_2$)

than the eutrophic reservoirs and less impacted upstream reaches. More field measurements of $pCO_2$, together with accurate flux calculations based on river-specific model parameters, are urgently required to provide more accurate estimates of GHG emissions from the Asian rivers that are now underrepresented in the global C budgets. Researchers working on individual river systems need to be linked to collaborative research networks to facilitate global synthesis of local field data. These synthesis efforts, combined with conceptual and mathematical models, will contribute to a better understanding of how

anthropogenic perturbations in rapidly urbanizing watersheds across Asia and other continents enhance discontinuities in riverine metabolic processes and C fluxes and hence transform the 'natural' river assumed in the long-standing river continuum model to an 'anthropogenic' system.

**Key words:** Asian rivers, carbon dioxide outgassing, dissolved organic carbon, greenhouse gases, particulate organic carbon,

river impoundment, riverine carbon flux, urbanization, water pollution

## 1 Introduction

Inland waters play a pivotal role in the global carbon (C) cycle by storing, transporting, or transforming inorganic and organic C components along the hydrologic continuum linking the land and oceans (Kempe, 1982, 1984; Cole et al., 2007; Battin et al., 2009). Recent syntheses have provided both revised estimates for the riverine transport of dissolved organic C

(DOC) and particulate organic C (POC) and much higher estimates for the exchange of $CO_2$ between the atmosphere and inland waters than the previous estimates (Raymond et al., 2013; Regnier et al., 2013; Wehrli, 2013; Ward et al., 2017). Monitoring data are sparse for many river systems in Asia and Africa, leaving many blind spots in global syntheses of riverine C transport and emission. Despite some ongoing efforts to build global river chemistry databases such as GLORICH (Hartmann et al., 2014), the paucity and spatial inequality in monitoring data are limiting our capacity to provide spatially

explicit estimates for the riverine C transport and emission. Specifically, C released from anthropogenic sources in rapidly urbanizing watersheds around the world increases the uncertainty of the current global riverine C flux estimates (Regnier et al., 2013). Concurrent anthropogenic perturbations to the river systems, including eutrophication, altered sediment regimes, and increased water residence time in impounded rivers, can significantly change the riverine processing of organic matter (OM) and greenhouse gas (GHG) outgassing (Stanley et al., 2012; Regnier et al., 2013; Crawford et al., 2016). Compared to

the mobilization of anthropogenic C associated with erosion and weathering, much less is known about the amount, quality, and spatially explicit distribution of C released from sewage and urban non-point sources (Griffith et al., 2009; Bhatt et al.,



2014; Butman et al., 2015). Although enhanced lability and mineralization of organic C have been observed in streams and rivers draining urbanized watersheds (Hosen et al., 2014; Kaushal et al., 2014), only a few studies have linked the altered chemical composition and lability of urban runoff and wastewater treatment plant (WWTF) effluents to organic C

mineralization and $CO_2$ evasion in rivers and estuaries (Griffith and Raymond, 2011; Wang et al., 2017; Yoon et al., 2017). Asian rivers have been estimated to account for up to 40–50% of the global inorganic and organic C fluxes from the land to the oceans (Degens et al., 1991; Ludwig et al., 1996; Schlünz and Schneider, 2000; Dai et al., 2012). However, the lack of high quality data and poor spatial coverage have constrained our ability to estimate the contributions of Asian river systems to the global riverine C fluxes in general and $CO_2$ outgassing in particular (Schlünz and Schneider, 2000; Lauerwald et al.,

2015; Li and Bush, 2015). For instance, obtaining $pCO_2$ data measured in Southeast Asian rivers was suggested as a top priority to reduce the large uncertainty in estimating the global riverine $CO_2$ outgassing (Lauerwald et al., 2015). Moreover, little is known about organic C export and $CO_2$ outgassing from streams and rivers draining rapidly urbanizing watersheds in developing Asian countries (Bhatt et al. 2014). A few recent studies conducted in the metropolitan areas of China and Korea have suggested that GHG emissions from polluted waterways in large urbanized watersheds may be underappreciated as

sources of GHGs (Wang et al., 2017; Yoon et al., 2017). Despite recent booms in the construction of large dams in many large rivers across the region, little attention has been paid to impoundment effects on GHG emissions (Chen et al., 2009; Hu and Cheng, 2013). Considering the role of dams in storing huge amounts of sediment and organic C (Syvitski et al., 2005; Maavara et al., 2017), a mechanistic understanding of GHG emissions from impounded rivers can provide more insights into the anthropogenic perturbations to the C fluxes of the dammed river systems.

This review aimed to provide a latest update on major anthropogenic perturbations affecting the riverine C fluxes in Asian river systems, focusing on the impacts of water pollution and river impoundments on riverine $CO_2$ dynamics in South (S), Southeast (SE), and East (E) Asian regions that account for the largest fraction of river discharge and C exports from Asia and Oceania (Fig. 1). We compared reported values of $pCO_2$ either measured or estimated for major river basins in three Asian regions, including the Ganges, the Mekong, and the Yellow River as models systems to assess the current status. We

also compared $pCO_2$ values among different components of the river basin (mainstem, headwater, tributary, and impoundment) to examine how water pollution and impoundments alter the riverine metabolic processes and $CO_2$ emissions. Many of the reported values were estimated from pH and alkalinity data available from the literature and water quality databases such as GLORICH (Global River Chemistry Database; Hartmann et al., 2014). Considering potential overestimations of water $pCO_2$ associated with organic acid contributions and increased sensitivity to alkalinity in acidic,

organic-rich waters with low carbonate buffering (Abril et al., 2015), we provided methodological details for the calculated $pCO_2$ values, if the cited references considered these pH and alkalinity effects. Another important goal was to integrate various concepts of riverine biogeochemical processes into a conceptual framework for assessing human impacts on the riverine C fluxes in human-modified river systems. Given the pace and wide-ranging impacts of urbanization and river impoundments, the traditional view of the river continuum developed for natural streams and rivers (Vannote et al., 1980)

needs to be revised to incorporate altered regimes of riverine metabolic processes and material fluxes. This review and



ensuing synthesis efforts are expected to provide scientifically more robust conceptual frameworks to understand and predict how human-induced perturbations in rapidly urbanizing watersheds across Asia transform riverine metabolic processes and C fluxes away from the 'natural' states assumed in the traditional river continuum model.

## 2 Why do Asian rivers matter in the global C cycle?

Global syntheses of riverine C fluxes have been focused on large rivers for which C flux data are available (e.g., Degens et al., 1991; Ludwig et al., 1996). We referred to these previous syntheses and a more recent synthesis of global river discharge (Milliman and Farnsworth, 2011) to scope the geographical extent of Asian river systems. We followed the continental categories used by Milliman and Farnsworth (2011), namely Asia and Oceania demarcated on Fig. 1, but did not consider Arctic rivers in Russia and rivers in Australia and New Zealand. This review focuses on S, SE, and E Asian regions where

river systems are commonly affected by increasing human impacts and for which data are available to address major review themes. Many large rivers in three Asian regions (Fig. 1) belong to either 34 global rivers with basin areas greater than 500,000 km$^2$ (Milliman and Farnsworth, 2011) or 32 global rivers included in the global synthesis of riverine C fluxes (Ludwig et al., 1996). The river systems in these Asian regions share some common hydrologic and demographic features (Table 1), including a large seasonality in discharge, high population densities (80–513 km$^{-2}$) compared to the global mean

(70 km$^{-2}$), and the wide range of per-capita annual discharge (23–8,594 m$^3$ yr$^{-1}$ person$^{-1}$ vs. the global mean: 4,901 m$^3$ yr$^{-1}$ person$^{-1}$).

Rivers draining the Asian continent were estimated to account for up to 35%, 50%, and 39% of the global discharge, total organic C (TOC) export, and dissolved inorganic C (DIC) export (Degens et al., 1991). Schlünz and Schneider (2000) provided a lower estimate for TOC export by Asian rivers (175. 2 Tg C yr$^{-1}$; 40% of the global TOC flux). Ludwig et al.

(1996) provided separate estimates for the export of DOC (69.01 Tg C yr$^{-1}$) and POC (76.40 Tg C yr$^{-1}$) by Asian rivers, which represented 34% and 44% of the corresponding global fluxes, respectively. As indicated by the distinctively lower ratio of DOC to POC (0.9) compared to other regions with the ratio exceeding 1, many Asian rivers draining the erosion-prone mountainous terrain deliver more POC than DOC, particularly during the monsoon period (Ittekkot et al., 1988; Ludwig et al., 1996). More recent syntheses of the published data have corroborated the quantitative importance of Asian

rivers in the global fluvial C fluxes (Dai et al., 2012; Huang et al., 2012; Galy et al., 2015).

The previous synthesis efforts heavily depended on a small number of data sets that were collected in several large Asian rivers during the 1970s and 1980s. Therefore, these data are limited in assessing how recent environmental changes taking place across Asian river systems have been altering riverine C fluxes. There have been few systematic assessments of effects of impoundments and water pollution on the C fluxes of Asian rivers (Sarma et al., 2011; Ran et al., 2014; Li and Bush,

2016). For example, a cascade of dams constructed along the upper Mekong River since the 1990s have been implicated to cause a wide range of downstream impacts including decreases in water and sediment flow (Li and Bush, 2016). Although it is expected that declining sediment flux can significantly alter POC and associated C fractions along downstream reaches, little is known about impoundment effects on the fluxes of POC, DOC, DIC, and $CO_2$ in the upper and lower Mekong River.



This lag between the real-time environmental changes and scientific assessments based on outdated data is quite surprising,

given the magnitude and pace of the environmental changes occurring across Asia. A recent report on the riverine export of plastic debris illustrated alarming trends of environmental changes occurring in Asian river systems by showing that 8 of top 10 rivers exporting the highest loads of plastics to the oceans drain the rapidly urbanizing watersheds across Asia (Schmidt et al., 2017).

An important factor affecting the riverine C fluxes in Southeast Asia is the rapid land use change including deforestation and

associated peatland drainage, which can enhance C mobilization from the coastal peatlands and organic rich soils in the region (Baum et al., 2007; Wit et al., 2015). A detailed study on rivers draining disturbed peatland areas in Indonesia and Malaysia revealed $p$CO$_2$ values in the low-salinity areas between 4000 and > 8000 µatm. Despite these relatively high pCO2 values, the rates of CO$_2$ outgassing were found to be lower than what global models have generally suggested, implying that local-scale processes and sources of C need to be better constrained to improve region-specific C budgets (Wit et al., 2015).

Nevertheless, land use change seems to be a critical factor affecting CO$_2$ outgassing from the rivers of Southeast Asian islands and lowland areas (Wit et al., 2015).

## 3 Conceptual framework for understanding interactive effects of changing land-water-scape and climate on riverine C fluxes

Human-induced land changes, as manifested in agricultural lands and urban areas, drive changes in biogeochemical cycles and climates, with altered terrestrial biogeochemical cycles often leading to pollution in downstream aquatic systems (Grimm et al., 2008). As a consequence of global urbanization, human influences are pervasive across the interacting terrestrial and aquatic patches of riverine landscapes or riverscapes (Allan, 2004; McCluney et al., 2014). To emphasize dominant human influences on connectivity and interactions among terrestrial and aquatic patches of the riverine networks,

we term these anthropogenically modified riverscapes "anthropogenic land-water-scapes". Compared to the previous use of the term "land-waterscape" focusing on terrestrial-aquatic boundary conditions in urbanized watersheds (Cadenasso et al., 2008), our use is more general and inclusive, covering longitudinal linkages between less or more modified up- or downstream reaches. Rivers dominated by effluents of wastewater treatment plants (WWTPs) in semiarid regions illustrate how human activities, through water withdrawal and wastewater generation in this specific case, modify flows of water and

materials across the anthropogenic land-water-scape, affecting concomitantly hydroclimates and aquatic ecosystems along downstream reaches (Luthy et al., 2015). Hydro-biogeochemical cycles across the components of these anthropogenic land-water-scapes often display idiosyncratic features unique to urbanized systems, which cannot be explained or predicted by traditional concepts and models built for natural or minimally impacted systems (Allan, 2004; Hutyra et al., 2014).

The concept of the river continuum from headwaters to mouth has been a powerful tool to represent longitudinal

connectivity in river ecosystem structure and function over the last five decades (Vannote et al., 1980; Webster, 2007), Gradual, but continual latitudinal changes in OM composition and metabolic rates envisaged in the original river continuum




concept (Vannote et al., 1980; Fig. 2) have resulted in a wide range of ramifications into studies exploring various structural and functional aspects of fluvial systems, including those of OM chemical diversity (Mosher et al., 2015) and biodegradability (Catalán et al., 2016), and $CO_2$ outgassing (Hotchkiss et al., 2015). A prevailing idea underlying these

approaches has been the selective degradation of labile components of OM during transit across the continuum, which has been successful in explaining the critical role of water retention time for the downstream evolution of the composition and biodegradability of DOM in the river systems with a relatively high proportion of natural lakes and/or low levels of anthropogenic perturbations (Koehler et al., 2012; Weyhenmeyer et al., 2012; Mosher et al., 2015; Catalán et al., 2016). According to the reactivity continuum model, the composition of DOM gradually becomes dominated by highly degraded

compounds as a result of prolonged exposure of DOM to biodegradation, resulting in a downstream continuum of DOM reactivity (Koehler et al., 2012; Catalán et al., 2016).

Despite the wide use of the river continuum as a conceptual framework for understanding various riverine processes, it has been criticized for overlooking an increasingly recognized reality that specific rivers are often divided into discrete segments that are hierarchically nested in a river network (Townsend, 1996; Poole, 2002). Discrete segments along a river network can

occur as a result of "abrupt transitions between adjacent segments with dissimilar physical structure" within the hierarchically nested river network (Poole, 2002). These abrupt transitions between discrete segments can occur temporarily, as illustrated by seasonal variations in water connectivity (Casas-Ruiz et al., 2016). As depicted in Fig. 2, examples of human-induced discontinuities include those created by dams in regulated rivers (Ward and Standford, 1983) and pollution-induced perturbations of the production-respiration balance in the eutrophic river (Kempe, 1984; Garnier and Billen, 2007).

Although the river continuum concept was originally proposed to provide a template for integrating physical environments and biological processes of "natural, unperturbed stream ecosystems" (Vannote et al., 1980), discontinuities in fluvial processes and biogeochemical fluxes can be accentuated by anthropogenic perturbations.

River impoundments and water withdrawal alter not only the rates of runoff and sediment transport (Syvitski et al., 2005) but also aquatic primary production and its effects on OM biodegradation (Stanley et al., 2012). Although rivers tend to reset

physical and ecological conditions toward the pre-disturbance state, river impoundment alters those conditions drastically along the distance up- or downstream from the dam termed 'discontinuity distance' (Ward and Standford, 1983, Stanford and Ward, 2001). This discontinuity distance was originally proposed as part of the 'serial discontinuity concept' that attempted to provide a theoretical perspective of the river systems regulated by dams (Ward and Stanford, 1983). According to this concept, stream regulation by dams can induce disturbances to the gradual processes envisaged in the river continuum

concept, shifting a given physical or biological parameter longitudinally (Ward and Stanford, 1983). For example, Vannote et al. (1980) envisaged that parameters such as the ratio of production to respiration (P/R) and diel temperature difference (ΔT) would exhibit a specific longitudinal pattern shown in Fig. 2. Stream regulation can shift this longitudinal patter along the discontinuity distance. Although serial discontinuity concept has been a useful conceptual framework for assessing human influences in the regulated lotic systems, its presuppositions including no disturbances other than impoundment

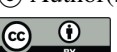



(Ward and Stanford, 1983) limit its use to urbanized river systems with other man-made structures and high levels of eutrophication associated with sewage and urban runoff.

Kaushal and Belt (2012) proposed an urban watershed continuum framework that recognizes a continuum of engineered and natural hydrologic flowpaths across the urbanized watershed. In the sense of spatial disconnection within the hierarchically nested river network (Poole, 2002), this urban watershed continuum is actually "a continuum with discontinuities" (*sensu*

Poole, 2002), in which the natural land-water hydrologic connectivity common in low-order streams is replaced by urban structures such as sewers and stormwater drains. The lateral transfer of water and associated materials via networks of engineered urban structures not only creates departures from the natural patterns or "discontinuities" in the hydrologic paths across the terrestrial-aquatic interface, but also exert extraordinary impacts on downstream fluxes and transformations of OM and nutrients (Paul and Meyer, 2001; Allan, 2004; Garnier and Billen, 2007; Lookingbill et al., 2009; Kaushal and Belt,

2012). By saying "the valley rules the stream", Hynes (1975) emphasized the importance of the terrestrial-aquatic connectivity in fluvial systems. In human-modified river systems, the valley is often separated from the stream or replaced by engineered structures that release pulses of water and materials, creating abrupt transitions across the land-water interface and stream segments (Fig. 2). As depicted in Fig. 2, urban structures across the land-water interface can also result in pulsatile flows of water and materials as a combined consequence of increased runoff from the impervious urban surface and

discharges from WWTPs, stormwater drainages, and combined sewer overflows (Paul and Meyer, 2001; Garnier and Billen, 2007; Kaushal and Belt, 2012). Although wastewater can bring pulses of OM and nutrients to the receiving urban water systems, its impact on riverine metabolism and $CO_2$ outgassing has not been investigated in many Asian river systems except for a few exploratory studies (Guo et al., 2014; Yoon et al., 2017).

Human-modified river networks often lack dynamic movements and flow adjustment and are therefore limited in their ability

to buffer against disturbances such as floods and water stress (Palmer et al., 2008). Climate models have suggested that perturbations in the global water cycle accompanying climatic warming can increase river discharge in many parts of the world (Milly et al., 2005). Although increases in river discharge have been detected for some large basins over the last century (Labat et al., 2004), globally no consistent pattern has been established. For example, cumulative discharge from many mid-latitude rivers including rivers draining arid regions of Asia have decreased substantially as a result of concurrent

changes in precipitation and anthropogenic perturbations such as damming, irrigation, and inter-basin water transfers (Milliman et al., 2008). While discharge from most of Asian rivers except Siberian rivers and the Brahmaputra has declined, the most striking decrease exceeding –50% was observed for the Indus and Yellow River (Milliman et al., 2008). Although the observed decreases in the discharge of many Asian rivers can be largely attributed to increased river impoundment and diversion, we also need to consider year-to-year variations in the strength of the monsoon system that result in a large

seasonality in river discharge and transport of sediment and C. In this context, it is an important proactive effort to predict how riverine C fluxes would respond to future changes in monsoon rainfall patterns.



## 4 Effects of river impoundments

According to a recent estimate based on the Global Reservoir and Dam database (GRanD), there may exist about 16.7
million reservoirs larger than 0.01 ha globally, with a combined storage capacity of 8069 km$^3$ (Lehner et al., 2011). Out of
6862 dams registered in GRanD with a total storage capacity of 6197 km$^3$, 1906 dams located in Asia (excluding Middle
East Asia and western Russia) store 1625 km$^3$ water, accounting for 26% of the global storage capacity. AQUASTAT, a
global water information system operated by Food and Agriculture Organization (FAO), provides a similar estimate for the
reservoir storage capacity of southern and eastern Asia: 1325 km$^3$ (Table 2; FAO, AQUASTAT). Over the last decades,
rivers across Asia have been increasingly impounded by dams of various type and size and a recent boom in constructing
hydroelectric dams is posing an unprecedented challenge for the sustainable management of the affected river basins
(Grumbine et al., 2012; Winemiller et al., 2016). River impoundments not only affect downstream flows but also disrupt the
ecological and biogeochemical connectivity of rivers (Lehner et al., 2011; Winemiller et al., 2016; Maavara et al., 2017). A
growing number of dams have been decreasing both water and sediment fluxes, sequestering over 100 billion metric tons of
sediment and 1 to 3 billion metric tons of C in reservoirs constructed over the last 50 years (Syvitski et al., 2005). Many
Asian and African rivers have been among the most affected river systems, with greatly reduced sediment exports to the
oceans compared to the pre-dam era (Syvitski et al., 2005).

The Yellow River in northern China is an excellent example illustrating basin-scale anthropogenic perturbations to the
sediment and C transport (Fig. 3). While the sediment load peaked during the period 1800−1950 following a millennium of
aggravating soil erosion in the Loess Plateau, it has drastically declined by 90% over the last 60 years resulting primarily
from human activity (Chen et al., 2015; Wang et al., 2015). Particularly, silt check dams and reservoirs, along with hillslope
soil conservation measures such as terrace farming, played a crucial role in the observed reductions. More than 110,000 silt
check dams have been constructed in the Loess Plateau since the 1950s, trapping approximately 21 billion tons of sediment
in the reservoirs (Zhang et al., 2016). A conservative estimate indicates that the rate of annual POC trapping in more than
3,000 large dams (excluding 110,000 silt check dams) within the Yellow River basin can amount to 3.3–4.3 Tg C yr$^{-1}$ (1 Tg
$=10^{12}$ g; Zhang et al., 2013; Ran et al., 2014), which is similar in magnitude to the total organic C export to the Bohai Sea
(4.1 Tg C yr$^{-1}$; Ran et al., 2014; Fig. 3). Furthermore, as an important attempt to control soil erosion, the Chinese
government initiated the largest ever revegetation program in history called the "Grain for Green Project" from the late
1990s. Implementation of this project has made an additional contribution to the decreasing trend of sediment flux since then
(Wang et al., 2015), and the resulting impacts on soil organic C stocks and the riverine C fluxes are also far-reaching (Feng
et al., 2013). It is anticipated that human activity will further affect sediment and C dynamics in the Yellow River basin in
the foreseeable future. For example, another 163,000 silt check dams are being planned on the Loess Plateau through 2020
(Zhang et al., 2016), and the resulting impact on OC burial will be substantial.

While POC trapped in the reservoir sediments has been considered as a relatively stable C sink (Zhang et al., 2013), little is
known about the transformations between POC and DOC and the release of $CO_2$ and other GHGs during the biodegradation



of POC and DOC in the increasingly impounded rivers across Asia (Sarma et al., 2011). As illustrated by the relatively high rate (27%) of $CO_2$ emission from the POC eroded from the entire Yellow River basin (Ran et al., 2014; Fig. 3), the organic C in highly turbid waters of many Asian rivers can become an important source of $CO_2$ during the fluvial transport and sediment storage. Although the rate of $CO_2$ evasion from the water surface can increase in the impounded river reaches as

longer residence times tend to create favourable environments for microbial biodegradation of organic C (Ittekkot et al., 1985; Ran et al., 2015a), the countering effect of enhanced planktonic uptake of $CO_2$ in the euphotic reservoir surface has rarely been compared against biodegradation (refer to the wide range of $p CO_2$ summarized in Table 3). Based on extensive $CO_2$ evasion measurements in the river-reservoir-river continuum on the Loess Plateau, Ran et al. (2017a) found that the Loess Plateau reservoirs acted as relatively small sources or even sinks of C, due largely to the significantly reduced

turbulence and enhanced photosynthesis. In comparison, both the upstream and downstream reaches were larger C sources for the atmosphere driven by strong turbulence in the aqueous boundary layer. Similarly, Liu et al. (2016) observed over 60% decreases in $p CO_2$ along the eutrophic impounded reaches of the Three Gorge Reservoir in the Yangtze River, but they suggested that the effect of enhanced primary production would be rather temporary and local compared to the predominant role of allochthonous C in fuelling $p CO_2$ dynamics.

In the Lancang River (the upper Mekong River located in China), Li et al. (2013) observed dramatic increases in the abundance of phytoplankton and a shift of the algal community toward *Chlorophyta* and *Cyanophyceae*, following the construction of cascade dams since 1995. In a recent study that compared the emission rates of $CO_2$ and $CH_4$ across the upper riverine reach and six cascade dams along the Lancang River, Shi et al. (2017) found that gas emission rates, particularly those of $CH_4$, were highest in the most upstream and second newest dam and that % organic C in the reservoir

bottom sediment had decreased with the increasing age of the dams. These results suggest that favourable conditions created by river impoundments, such as increased water temperature and retention time, can stimulate OM processing in both the impounded water and trapped sediments, at least in the short term following the construction of dams on the high-POC mountainous rivers such as the Lancang.

The export of sediment and POC to the coastal waters surrounding the Indian subcontinent generally peaks during the

monsoon period from June to September when southwest monsoonal rainfalls drastically increase runoff (Sarma et al., 2012; Galy et al., 2013; Krishna et al., 2015a,b). Most of the major Indian rivers have been impounded by dams of various type and size to meet domestic and industrial water demands during the dry period. In the case of the Krishna river basin, dam construction has decreased the sediment load from 67.7 Tg yr$^{-1}$ measured at the upper reach to 4.11 Tg yr$^{-1}$ at the mouth of the river (Ramesh and Subramanian, 1988). Indian monsoonal rivers discharge the largest share of POC export from the

Asian rivers (Ludwig et al., 1996; Galy et al., 2015). Although altered discharges of these rivers as a consequence of climate variability and anthropogenic perturbations can have significant impacts on the amount and lability of DOC and POC exported to the Indian coastal waters (Ittekkot et al., 1985; Krishna et al., 2015a), the effects of large dams have been studied only in a few estuaries of the dammed rivers such as the Godavari estuary (Sarma et al., 2011). There has been no systematic investigation of altered rates of GHG emissions from the impounded reaches of the Ganges River (Table 3), although many



large dams and barrages have been constructed on the mainstem and major tributaries. Because of this poor spatial coverage of available field measurements, estimating GHG emissions from impounded rivers in India would be really challenging. Despite this lack of data, Panneer Selvam et al. (2014) ventured to estimate the emission rates of $CO_2$ and $CH_4$ for the entire India's inland waters at 22.0 Tg $CO_2$ yr$^{-1}$ and 2.1 Tg $CH_4$ yr$^{-1}$, respectively, by extrapolating their flux measurements at 45 water bodies in South India. While they provided the emission rates of 2.37 Tg $CO_2$ yr$^{-1}$ and 0.33 Tg $CH_4$ yr$^{-1}$ for the man-made reservoirs and barrages, a follow-up study (Li and Bush, 2015), which considered additional literature data from the large rivers in northern India and the estimates for reservoir downstream fluxes through spillways and turbines, proposed much higher estimates: 3.08 Tg $CO_2$ yr$^{-1}$ and 6.27 Tg $CH_4$ yr$^{-1}$.

A significant reduction in the primary production observed in the Godavari River, the largest monsoonal river in India, was ascribed to the removal of nitrogen and phosphorus in several dams (Das, 2000). It has been observed that the river delivers large amounts of terrestrial OM (mainly from C3 plants) to and supports heterotrophic activity in the estuaries and coastal waters (Sarma et al., 2009; 2014). Studies conducted in the Godavari estuary have found that during the peak discharge periods in the monsoon season, the estuary receiving discharge waters from an upstream dam exhibited extraordinarily high levels of $pCO_2$ up to 33,000 µatm compared to the dry season values lower than 500 µatm, presumably due to enhanced bacterial decomposition of the organic C released from the upstream dam in the highly eutrophic estuary (Sarma et al., 2011, Prasad et al., 2013). It demands further research to establish whether the monsoonal eutrophication in impounded reaches of Indian rivers usually leads to an enhanced heterotrophy, rather than a boost in autotrophy, increasing $CO_2$ outgassing from the impounded reaches and downstream estuaries (Prasad et al., 2013). Pradhan et al (2014) reported that during dry years lower rainfalls, combined with impoundment effects, increased the fraction of labile OM derived from the riverine and estuarine phytoplankton. Ramesh et al (2015) observed that damming of rivers and construction of reservoirs significantly increased the retention of particulate OM in the Godavari and Krishna rivers.

With regard to the fate of POC trapped in the reservoirs, an important question remains as to how the amount and lability of the stored C change in the short period following the dam construction (Barros et al., 2011; Maavara et al., 2017). Although large pulses of GHGs are released from the flooded vegetation and soil OM during the initial flooding phase (Abril et al., 2005; Chen et al., 2009; Hu and Cheng, 2013; Deshmukh et al., 2016, 2017), sedimentation can accumulate a growing amount of C in reservoir sediments, greatly decreasing the rate of $CO_2$ release from aging reservoirs (Barros et al., 2011). Large pulse emissions of $CO_2$ and $CH_4$ have been measured in the years following the construction of the Three Gorges Dam on the Yangtze River (Chen et al., 2009) and the Nam Theun 2 on the large tributary feeding into the middle reach of the Mekong River (Deshmukh et al., 2016, 2017), emphasizing the initial flooding phase as the hot moment of GHG emissions from the impounded river. A recent report on drought-enhanced emissions of GHGs in an old hydroelectric reservoir in Korea suggested that stochastic emissions during extreme climatic events can reverse the trend of declining C emissions from aging reservoirs by the offsetting effect of extreme events on the C accumulated in reservoir sediments over time scales of years to decades (Jin et al., 2016).





## 5 Effects of increasing water pollution in Asian river systems

Many streams and rivers across Asia are highly polluted by agricultural runoff and domestic and industrial wastewater, with water quality often exhibiting large seasonal variations associated with regional monsoon rainfall regimes (Park et al., 2010; Park et al., 2011; Evans et al., 2012, Bhatt et al., 2014). Using AQUASTAT data, Evans et al. (2012) estimated the annual wastewater generation in Asia around the year 2000 at 142 km$^3$, of which only an estimated 33–35% was treated before being discharged to streams and rivers. We used the latest data available on the AQUASTAT webpage to provide more up-

to-date estimates of water withdrawal and wastewater production, focusing on southern and eastern Asia (FAO, AQUASTAT). In 2010, the annual municipal water withdrawal in 24 southern and eastern Asian countries was 201.6 km$^3$, accounting for 43.4% of the global municipal withdrawal (464.1 km$^3$; Table 2). The total volume of the municipal wastewater generated each year within urban areas of these countries was 120.2 km$^3$. The generated wastewater included domestic, commercial, and industrial effluents, and storm water runoff, accounting for 38.6% of the global municipal

wastewater production (311.6 km$^3$). Based on the AQUASTAT and other published data, Mateo-Sagasta et al. (2015) estimated that each year more than 330 km$^3$ of the municipal wastewater are produced globally. Although the volume of the municipal wastewater generated in these Asian regions constitutes only ~0.9% of the renewable surface water available in these regions (14,027.1 km$^3$), both treated and untreated wastewater can have disproportionately large impacts not only on the water quality and ecological integrity of downstream aquatic ecosystems (Meybeck and Helmer, 1989; Evans et al., 2012)

but also on the riverine GHG emissions (Yoon et al., 2017).

Compared to the extensive studies conducted in polluted rivers and estuaries in Europe and North America (Frankignoulle et al., 1998; Borges et al., 2006; Hartmann et al., 2007; Borges and Abril, 2011; Griffith and Raymond, 2011; Amann et al., 2012; Joesoef et al., 2015), relatively few efforts have been made to measure $p$CO$_2$ in polluted Asian rivers, except for some large rivers and estuaries in East Asia (Zhai et al., 2005; Chou et al., 2013; Ran et al., 2015b; Yoon et al., 2017). These

studies, together with a small number of studies that used water chemistry data to estimate the levels of $p$CO$_2$ in major Asian rivers such as the Mekong (Li et al., 2013), the Yangtze (Ran et al., 2017b), the Ganges-Brahmaputra (Manaka et al., 2015), and Indian estuaries (Gupta et al., 2009; Sarma et al., 2012), underscored the importance of anthropogenic OM and nutrients for riverine CO$_2$ dynamics, particularly along the lower river reaches and estuaries draining highly populated areas. When published data of $p$CO$_2$ were compared between the headwaters and the tributaries feeding into the middle and lower reaches

of the major Asian rivers, tributary $p$CO$_2$ levels (mean: 2,128 µatm) tended be higher than those for the headwaters of the both global (mean: 1116 µatm) and Asian rivers (mean: 819 µatm) (Table 3 and references therein). Some studies have examined the effects of domestic and industrial wastewaters on the chemical composition and lability of riverine organic C (e.g., Guo et al., 2014). By comparing fluorescence excitation-emission matrices (EEMs) of DOM between the branches and tributaries of the Yangtze River estuary, Guo et al. (2014) found that labile DOM components delivered by the Huangpu

River, a highly polluted tributary, exerted a disproportionately large influence on the biodegradability of DOM in the Yangtze estuary. Direct underway measurements of $p$CO$_2$ along the river and estuarine reaches of a few river systems in



China also indicated a potential activation of riverine microbial processing and enhanced $CO_2$ evasion from polluted waterways (Zhai et al., 2005; Chou et al., 2013; Wang et al., 2017).

As eutrophication becomes more common in rivers draining highly populated watersheds as a consequence of global urbanization and water pollution, eutrophic rivers often exhibit metabolic rates that have rarely been observed in the flowing water systems under minimal to low human influences (Meybeck and Helmer, 1989; Hilton et al, 2006; Garnier and Billen, 2007). A recent report on $CO_2$ outgassing from a highly urbanized river in Korea suggested a potential regime shift in riverine metabolic processes by showing a shift in the relationship between Chl $a$ and $p$CO$_2$ from the upstream reach less enriched in nutrients and $CO_2$ to the eutrophic downstream reach receiving highly polluted urban tributaries carrying WWTP

effluents (Yoon et al., 2017; Fig. 5). As suggested by this case study that was conducted in the Han River where the middle reach is affected by cascade dams and the lower reach receives urban streams draining the Seoul metropolitan area, river eutrophication, in combination with impoundment effects, creates discontinuities in the metabolic processes along the longitudinally connected river reaches. In accordance with the findings of large spatial and seasonal variations in the balance between autotrophy and heterotrophy in eutrophic European rivers (Garnier and Billen, 2007), enhanced bacterial

degradation of OM of both allochthonous and autochthonous origin in the eutrophic lower reach receiving high loads of organic pollutants might greatly increase the level of $p$CO$_2$ despite the longitudinal increase in primary production with widening channel toward the river mouth, moving the regime of riverine metabolism away from those found in the less eutrophic upstream reach.

With the basin-wide average $p$CO$_2$ around 2,257 (147–9,659) µatm, the Yellow River water has been evaluated as a source

of $CO_2$ for the atmosphere (Ran et al., 2015a, b; Ran et al., 2017a; Table 3; Fig. 4). The $CO_2$ evasion from the fluvial network of the whole basin was estimated at 4.7–7.9 Tg yr$^{-1}$, with >70% emitting from the tributaries draining the Loess Plateau (Ran et al., 2014, 2015b; Fig. 3). Water pollution within the Yellow River basin in recent decades has become an increasingly pressing environmental issue for watershed management and water resources exploitation, in particular for the middle and lower reaches flanked by large industrial complexes and irrigated farmlands (Zhang et al., 2013). Agricultural

runoff, domestic sewage, and industrial wastewater released from the basin inhabited by a population > 100 million have been aggravating water pollution across the basin (Li et al., 2006; Lu et al., 2015). Anthropogenic pollutants may affect not only water quality but also the riverine C cycling within and beyond the basin. For example, drainage waters from agricultural croplands containing high loads of DOC have resulted in a significant increase in DOC concentration in the middle reach of the Yellow River, and wastewater discharged from large regional urban centers has greatly elevated the

concentrations of DOC and POC in the lower reach, particularly in winter (Zhang et al., 2013). In addition, recent studies focusing on polycyclic aromatic hydrocarbons (PAHs) and phthalic acid esters (PAEs) indicated extraordinarily high pollution levels, compared with rivers in other countries (Li et al., 2006; Sha et al., 2007). Tributaries typically exhibit higher pollution levels than the mainstem Yellow River, as a combined result of higher loads of pollutants discharged from local sources and the dilution in the mainstem by the higher flow. Relatively high POC concentrations in these highly polluted

river reaches (e.g., Li et al., 2006) suggest that increasing levels of water pollution can affect the riverine C cycling by





providing more sources of allochthonous C. There has been no systematic investigation of pollution impacts on the riverine transformations of organic C and GHG emissions in the Yellow River basin, although the potential biogeochemical impacts might be far-reaching beyond the basin to the downstream coastal ecosystems.

It is very difficult to evaluate overall pollution impacts on the mainstem Mekong River as very few systematic measurements
of the nutrient and C cycles of the Mekong River have been made, except for the short reaches of the lower Mekong in Laos and Cambodia (Alin et al., 2011; Ellis et al., 2012; Martin et al., 2013) and in the Mekong Delta (Borges et al., 2017). The basic properties of the Lower Mekong are consistent with the other big rivers of Southeast Asia and other similar climate zones. The mean annual DIC and alkalinity in the Mekong near Phnom Penh (~1.1 mM; J. Richey, unpublished data) was close to that of the Pearl River (1 mM), and intermediate between low-alkalinity rivers such as the Amazon (typical values in
the downstream reach at Obidos: 0.25 – 0.6 mM; Richey et al., 1990; Mayorga and Aufdenkampe, 2002) and high-alkalinity rivers such as the Mississippi and Yellow River (1.5 – 3 mM; Raymond and Cole, 2003; Ran et al., 2015a). The annual DIC flux of the Mekong was 3.95 Tg, with DIC and alkalinity both negatively correlated with discharge (J. Richey, unpublished data). This trend, which has also been observed in the Pearl (Zhang et al., 2007) and Yellow River (Ran et al., 2015a), has been attributed to the dominance of a weathering-based source in the dry season which is diluted by less ion-rich source
(rainwater) during the high-flow periods (Cai et al., 2008). The mean of reported values for the lower Mekong River is 1235 μatm (Table 3). Using a model based on pH and alkalinity, Li et al. (2013) reported a similar mean of $pCO_2$ for the lower Mekong River basin: 1090 (224–5,970) μatm. Alin et al. (2011) measured similar values (~1200 μatm) at eight mainstem locations in Laos and Cambodia. The seasonal trend in $pCO_2$ opposes the alkalinity and DIC trends, peaking in the flood season and lowest in the dry season, similar to several previous studies in tropical river systems (Sarma et al., 2011; Borges
et al., 2017).

The level of $pCO_2$ in the Mekong River tended to increase downstream, with an average of 812 μatm near Chiang Saen, Thailand increasing toward 1670 μatm in the Mekong Delta near Cantho, Vietnam (Li et al., 2013). The same study indicated potential effects of polluted tributaries on $CO_2$ emissions from the mainstem Mekong by showing relatively high $pCO_2$ values approaching 2000–3000 μatm in tributaries draining the highly populated areas such as the Tonle Sap near
Phnom Penh and local river channels along the Bassac in the Mekong Delta (Li et al., 2013; Table 3). In Phnom Penh, a combined drainage system delivers untreated municipal wastewater and storm runoff either directly or through four natural wetlands surrounding the city used for natural purification to the Tonle Sap and Mekong (Irvine et al. 2006). The poor purification capacity of the natural wetlands is too limited to counteract the ever increasing pollution levels of the municipal wastewater, resulting in a continuous deterioration of the water quality of wastewater-receiving rivers. MoWRAM (2016)
has recently reported that total nitrogen concentrations in the Mekong downstream of Phnom Penh increased by approximately 20% from 2012 to 2015. However, there has been no systematic assessments of local pollution impacts on the carbon fluxes of the Mekong. Measurements of $pCO_2$ along three freshwater channels in the Mekong Delta ranged between 1,895 and 2,664 μatm during the high flow periods of December 2003 and October 2004 (Borges et al., 2017) and exceeded

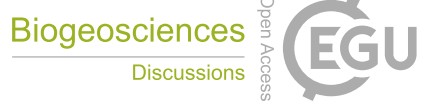

the range of 703–1,597 µatm observed in the upstream reach during the similar period (September–October, 2004 and 2005)
by Alin et al. (2011). As suggested by Borges et al. (2017), anthropogenic pollution sources in the densely populated and
cultivated areas of the Delta may release more $CO_2$ and biodegradable OM compared to the upstream reach.

As summarized in Table 3, many studies of aquatic $CO_2$ dynamics in India have been conducted in estuaries and coastal
areas (e.g., Mukhopadhyay et al., 2002; Biswas et al., 2004; Gupta et al., 2009; Sarma et al., 2012; Samanta et al., 2015),
except for the secondary data of $pCO_2$ that can be estimated based on $CO_2$ system calculations (Pierrot et al., 2006) and
water quality data collected in various headwaters (Sarin et al., 1989; Bickle et al., 2003; Chakrapani and Veizer, 2005) and
lower reaches (Manaka et al., 2015) of the Ganges-Brahmaputra. The values of $pCO_2$ estimated for some headwaters, lower
reaches, and tributaries of the Ganges basin (mean: 891; range: 16–2,778 µatm) were relatively low compared to other Asian
rivers (Table 3). Only one study directly measured the rates of $CO_2$ and $CH_4$ emission from various inland waters across
three southern Indian states (Panneer Selvam et al., 2014). In a study of human impacts on C dynamics in the Cochin estuary,
Southern India, Gupta et al. (2009) ascribed monsoonal $pCO_2$ increases up to 6,000 µatm to the enhanced decomposition of
the OM released from anthropogenic sources upstream. A particular attention has been paid to the emission of $CO_2$ and $CH_4$
from the Indian part of the deltaic region of the Ganges-Brahmaputra system that includes estuaries with contrasting
biogeochemical features: the anthropogenically impacted Hooghly estuary and the mangrove-dominated estuaries of
Sundarbans, the world's largest mangrove ecosystem (Mukhopadhyay et al., 2002; Biswas et al., 2004; Biswas et al., 2007;
Dutta et al., 2015, 2017; Samanta et al., 2015). The Hooghly estuary was found as net heterotrophic, with the fugacity of
$CO_2$ ($fCO_2$) and $CH_4$ levels varying from ~ 400–700 µatm and 10.3–59.25 nM, respectively (Mukhopadhyay et al., 2002;
Biswas et al., 2007). On an annual scale, the Hooghly estuary acts as a source of both $CO_2$ ($-2.78$ to $84.4$ mmol m$^{-2}$ d$^{-1}$;
Mukhopadhyay et al., 2002) and $CH_4$ (36.7–6193.5 nmol m$^{-2}$ hr$^{-1}$; Biswas et al., 2007) to the atmosphere. The estuaries of
Sundarbans were reported as being net heterotrophic, with $fCO_2$ and $CH_4$ concentrations varying between 160–797 µatm
(Biswas et al., 2004) and 54.20 ($\pm$ 5.06)–90.91 ($\pm$ 21.20) nM, respectively, and hence as functioning as a source for both $CO_2$
(314.6 mmol m$^{-2}$ d$^{-1}$) and $CH_4$ (4.66–17.59 µmol m$^{-2}$ d$^{-1}$) to the atmosphere (Dutta et al., 2015). Based on the mass balance
between source and sink inventories, Dutta et al. (2015) estimated that annually ~0.5 Gg of $CH_4$ was exported from the
estuaries of Sundarbans to the northern Bay of Bengal.) Although other components of the estuarine C biogeochemistry have
not yet been studied comprehensively in these systems, Samanta et al. (2015) reported a large annual DIC export [(3.1–3.7) x
$10^{12}$g] from the Hooghly estuary to the Bay of Bengal, suggesting estuarine conservative mixing as well as in situ
biogeochemical processes such as OM biodegradation and carbonate dissolution as key controls on the export and isotopic
composition of DIC. The saline estuarine zone functioned as an important source of DIC (up to 50% of the annual estuarine
export to the bay), particularly in the monsoon periods. Based on a few available data of wastewater discharge and DIC
concentrations, direct anthropogenic sources of DIC within the Hooghly basin were estimated to account for only 2–3% of
the river water DIC concentrations, although it remains unanswered how much the biodegradation of organic C released
from anthropogenic sources could contribute to DIC generation during the transport to the bay (Samanta et al., 2015).



Large knowledge gaps still exist in understanding human influences on emissions of $CO_2$ and other GHGs from the estuaries of the eastern India and associated river systems. There are some ongoing projects investigating the effect of anthropogenic pollution on the spatial variations of dissolved $CO_2$ and $CH_4$ in the Hooghly and Sundarbans estuaries (Kumar, unpublished

data). Both $CO_2$ (604–6524 μatm) and $CH_4$ (15.4–445.7 nM) varied over a wide range in the Hooghly compared to Sundarbans ($CO_2$: 332–490 μatm, $CH_4$: 41.6–71.5 nM). Estuarine $CH_4$ is completely exogenous with higher emission from the Hooghly compared to Sundarbans (Hooghly/Sundarbans = 1.55). $CO_2$ emission from the Hooghly is also significantly higher (Hooghly/Sundarbans = 247), suggesting a significant contribution of anthropogenic sources to the emission of GHGs from the polluted estuarine system such as the Hooghly. High-resolution monitoring conducted in other Indian estuaries and

adjacent coastal water bodies have revealed that decreased river discharge would create conditions for algal blooms that can lead to hypoxic conditions, biodiversity reduction, and decreased fish harvests (Sarma et al., 2009, 2010, 2011; Bharathi, 2015). In addition, decreased river discharge can enhance coastal upwelling, leading to further increase in $p$CO$_2$ levels and acidification in the adjacent coastal waters (Sarma et al., 2015).

**6 Summary and future research needs**

Despite local differences in prevailing environmental conditions and economic development, river systems across the eastern and southern Asia share two commonalities. First, large seasonal variations in precipitation and runoff associated with the Asian monsoon systems play a critical role in all hydrologically mediated riverine processes including those affecting C fluxes. Second, an unprecedented rapid growth in population and economy is causing tremendous perturbations to water and

material flows along the rivers that are not only regulated by dams and but also polluted by urban sewage and agricultural runoff. Given the large seasonality inherent in the monsoon hydrology, these anthropogenic land-water-scapes might be particularly vulnerable to climatic variability and extremes, as exemplified by strengthened flashy storm responses of sediment and C export from disturbed watersheds during extremely wet monsoon periods (Park et al., 2010; Jung et al., 2012). To better assess the interactive effects of climate change and human-induced perturbations to the riverine networks of

interacting land and water patches, we suggest that the long-standing concept of river continuum assuming 'natural' river states should be complemented with alternative perspectives of 'discontinuity' or 'discontinuous continuity' (*sensu* Poole, 2002) in riverine metabolisms and C fluxes created in impounded and/or eutrophic reaches of rivers draining increasingly urbanizing watersheds across Asia.

This review identified an alarming regional trend concerning dam construction booms and ensuing impoundment-induced

alterations of the rates of GHG emissions and sediment C storage in major rivers across the southern and eastern Asian regions. As illustrated by large pulse emissions of $CO_2$ and $CH_4$ in the years following the construction of the Three Gorges Dam on the Yangtze River and cascade dams and the Nam Theun 2 in the Mekong River basin, flooded soils and vegetation can become major sources of GHGs during the initial years following dam construction. Long-term changes in GHG emissions and sediment C storage might vary with dam location, initial conditions of the flooded area, and land use changes




occurring within the watersheds. As summarized in Table 3, there have been very few studies that measured or estimated $pCO_2$ in the impounded reaches of the dammed Asian rivers, making it very difficult to constrain factors crucial for the spatial and temporal variations in $pCO_2$ along the impounded reaches. The current booms of mega dam construction across Asia, which starkly contrast with very few large dam projects commissioned in Europe and North America over the recent decades, urgently deman more systematic assessments of far-reaching environmental impacts including riverine C transport

and GHG emissions. A relevant, but rarely investigated research topic is as to how more frequent droughts associated with regional climate change can reverse the gradually decreasing GHG emissions from the impounded river reaches through enhanced decomposition of the C stored in the reservoir bottom sediment.

Although rapid urbanization across Asia is aggravating eutrophication and organic pollution in many large rivers draining metropolitan areas with a limited capacity of wastewater treatment infrastructure, the scarcity of high-quality monitoring

data represents a huge challenge for a more thorough assessment of the current status of riverine metabolisms and $CO_2$ outgassing from the impacted rivers. Some exploratory studies conducted in highly urbanized watersheds in East Asia (e.g., Wang et al., 2017; Yoon et al., 2017) have questioned whether the conventional conceptual framework perceiving riverine C fluxes as a gradual longitudinal continuum can address large cross-scale variations and pulsatile patterns of riverine $CO_2$ outgassing observed in the highly modified river systems. Given the large share of the studied Asian regions (~40%) in the

global municipal wastewater production and disproportionately poor wastewater treatment infrastructure, it remains largely unknown whether our current understanding of biogeochemical processes in urban river systems in Europe and North America can help explain idiosyncratic features of OM composition and turnover in streams and rivers contaminated with high loads of nutrients and raw sewage. Building on conceptual and predictive models developed for highly eutrophic river systems in Europe and North America (e.g., Garnier and Billen, 2007), we need to develop new integrative frameworks to

explain river-specific responses to the unprecedented pace and scale of urbanization and water pollution.

In addition to the general regional patterns of river impoundments and eutrophication, there are also some other emerging research challenges. As observed in the Tibetan Plateau, urbanization and dam construction have expanded to the upstream headwater reaches of the large rivers such as the Mekong, Yangtze, and Yellow River. These anthropogenic perturbations, coupled with the fragile ecosystems on the Tibetan Plateau, are likely to further affect the riverine C cycle in the context of

climate change impacts. The magnitude of perturbations would be much larger than previously thought and the impact may be more widespread. Given the considerable C storage in soils and wetlands on the Tibetan Plateau, studying the human-induced C dynamics is of great importance for the global C cycle. In the case of Indian river systems, large-scale of modifications in river discharge and C fluxes are being expected as a consequence of a growing number of dams and a new national river development project called Inter-Linking of Rivers (Ministry of Water Resources, River Development, and

Ganga Rejuvenation; http://wrmin.nic.in/forms/list.aspx?lid=1279). Because the Inter-Linking Rivers project may have far-reaching impacts on the hydrology and biogeochemistry of Indian rivers and adjacent coastal waters, there is an urgent need to collect field data required to assess the current status and longer-term effects of the project.

To address both region-wide issues and emerging local challenges, more efforts should be given to build collaborative research networks that can provide practical guides and methodologies for designing and conducting field monitoring of 535 riverine C fluxes at multiple spatial and temporal scales. Although this review could not address in detail technical issues related to $CO_2$ measurements and flux calculation, an important, but very challenging task in providing more accurate estimates of $CO_2$ outgassing is obtaining more reliable $pCO_2$ measurements and gas transfer velocities required for $CO_2$ flux calculation (Raymond et al., 2012; Yoon et al., 2016). Most datasets used for assessing regional-scale $CO_2$ outgassing have employed scaling laws based on coefficients derived from relatively flat environments compared to the rather steep terrain 540 characteristic of many parts of Asia (c.f. Raymond et al., 2012). Region-specific scaling laws and coefficients, along with more $pCO_2$ measurements in many unexplored rivers, would contribute to reducing large uncertainties in current estimates of riverine C fluxes.

**Data availability**

Data are available and can be requested from the corresponding author (jhp@ewha.ac.kr).

**Author contribution**

All authors contributed to data acquisition, the discussion of concepts and research topics, and manuscript preparation. The manuscript was written through concerted efforts of all authors, coordinated by J.-H. Park.

**Competing interests**

The authors declare that they have no conflict of interest.

**Acknowledgements**

This work was supported by the Asia-Pacific Network for Global Change Research (CRRP2016-01MY-Park). JHP was also supported by an additional funding from the National Foundation of Korea (2017R1D1A1B06035179). We thank Dr. Tae Kyung Yoon for providing a figure modified from the supplementary information of his publication.



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





Table 1. Geographic and demographic features of the major river systems addressed in the review, in comparison with Asian and global sums.

| River | Receiving sea | Region | Basin area (10³ km²) | Annual discharge (km³ yr⁻¹) | Population (×10⁶) | Population density (per km²) | Annual discharge per capita (m³ yr⁻¹) |
|---|---|---|---|---|---|---|---|
| Ganges | Bay of Bengal | S Asia | 980 | 490 | 411 | 419 | 1193 |
| Brahmaputra | Bay of Bengal | S Asia | 670 | 630 | 145 | 216 | 4353 |
| Indus | Arabian Sea | S Asia | 980 | 5 (90) | 220 | 224 | 23 (410) |
| Krishna | Bay of Bengal | S Asia | 260 | 12 (62) | 101 | 390 | 118 (611) |
| Godavari | Bay of Bengal | S Asia | 310 | 92 (120) | 121 | 390 | 761 (993) |
| Mekong | South China Sea | SE Asia | 800 | 550 | 64 | 80 | 8594 |
| Yellow | Yellow Sea | E Asia | 750 | 15 (43) | 120 | 160 | 125 (358) |
| Yangtze | East China Sea | E Asia | 1800 | 900 | 475 | 264 | 1894 |
| Pearl River | South China Sea | E Asia | 490 | 260 | 95 | 193 | 2749 |
| Han River | Yellow Sea | E Asia | 25 | 17 | 13 | 513 | 1326 |
| Asia total | | | 32518 (32518) | 11000 (13196) | 4835 | 148 | 2274 |
| Global total | | | 105000 (106326) | 36000 (38170) | 7345 | 70 | 4901 |

River basin area and discharge data were obtained from Milliman and Farnsworth (2011), supplemented with Asian and global sums in parentheses from Ludwig et al., (1996). Pre-diversion discharge data are provided in parentheses for the rivers where discharge has substantially decreased in recent years because of river diversion, reservoir construction, and irrigation. The Asia total discharge provided by Milliman and Farnsworth (2011) was estimated for all rivers of Asia and Oceania, excluding Arctic rivers in Russia. Population data were taken from various sources including Schmidt et al. (2017) and CIA World Factbook (https://www.cia.gov/library/publications/the-world-factbook/index.html/; last accessed on 10 January 2018). Asia total population is the population for all countries belonging to Asia and Oceania excluding Russia.





Table 2. Summary of water use and wastewater production in southern and eastern Asia. Source of data: AQUASTAT, a global water information system operated by Food and Agriculture Organization (FAO; http://www.fao.org/nr/water/aquastat/sets/index.stm).

| Country | Population | Renewable surface water | Dam capacity | | Municipal water | | Municipal wastewater | | | |
|---|---|---|---|---|---|---|---|---|---|---|
| | | | | | | | Produced | | Treated | |
| | (2015) | (2014) | | | | | | | | |
| | $\times 10^3$ | $km^3\ yr^{-1}$ | $km^3$ | Year | $km^3\ yr^{-1}$ | Year | $km^3\ yr^{-1}$ | Year | $km^3\ yr^{-1}$ | Year |
| Bangladesh | 160996 | 1206.0 | 6.5 | 2013 | 3.6 | 2008 | 0.7 | 2000 | | |
| Bhutan | 775 | 78.0 | | | 0.0 | 2008 | 0.0 | 2000 | | |
| Brunei | 423 | 8.5 | 0.0 | 2010 | 0.2 | 2009 | | | | |
| Cambodia | 15578 | 471.5 | | | 0.1 | 2006 | | | 0.0 | 1994 |
| China | 1407306 | 2739.0 | 829.8 | 2013 | 75.0 | 2013 | 48.5 | 2013 | 49.3 | 2014 |
| North Korea | 25155 | 76.2 | 13.6 | 2015 | 0.9 | 2005 | | | | |
| India | 1311051 | 1869.0 | 224.0 | 2005 | 56.0 | 2010 | 15.5 | 2011 | 4.4 | 2011 |
| Indonesia | 257564 | 1973.0 | 23.0 | 2015 | 14.0 | 2005 | 14.3 | 2012 | | |
| Japan | 126573 | 420.0 | 29.0 | 1993 | 15.4 | 2009 | 16.9 | 2011 | 11.6 | 2011 |
| Laos | 6802 | 333.5 | 7.8 | 2010 | 0.1 | 2003 | 0.1 | 2008 | 0.0 | 1995 |
| Malaysia | 30331 | 566.0 | 22.5 | 2015 | 3.9 | 2005 | 4.2 | 2009 | 2.6 | 2009 |
| Mongolia | 2959 | 32.7 | 0.3 | 2015 | 0.1 | 2009 | 0.1 | 2012 | 0.1 | 2006 |
| Myanmar | 53897 | 1157.0 | 15.5 | 2005 | 3.3 | 2000 | | | 0.0 | 1995 |
| Nepal | 28514 | 210.2 | 0.1 | 2015 | | | | | 0.0 | 2006 |
| Pakistan | 188925 | 239.2 | 27.8 | 2015 | 9.7 | 2008 | 3.1 | 2011 | 0.0 | 2002 |
| Papua New Guinea | 7619 | 801.0 | 0.7 | 2010 | 0.2 | 2005 | | | | |
| Philippines | 100699 | 444.0 | 6.3 | 2006 | 6.2 | 2009 | 1.3 | 2011 | | |
| South Korea | 50293 | 67.1 | 16.2 | 1994 | 6.9 | 2005 | 7.8 | 2011 | 6.6 | 2011 |
| Singapore | 5619 | | 0.1 | 2015 | 1.1 | 2005 | 0.5 | 2013 | 0.5 | 2013 |
| Sri Lanka | 20715 | 52.0 | 5.9 | 1996 | 0.8 | 2005 | 0.1 | 2009 | | |
| Thailand | 67959 | 427.4 | 68.3 | 2010 | 2.7 | 2007 | 5.1 | 2012 | 1.2 | 2012 |
| Timor | 1185 | 8.1 | | | 0.1 | 2004 | | | | |
| Vietnam | 93448 | 847.7 | 28.0 | 2010 | 1.2 | 2005 | 2.0 | 2012 | 0.2 | 2012 |
| S/SE/E Asia | 3964386 | 14027.1 | 1325.2 | | 201.6 | | 120.2 | | 76.5 | |
| World | 7344837 | 52952.7 | 7039.6 | | 464.1 | | 311.6 | | 187.1 | |





Table 3. Summary of $p$CO$_2$ measured (M) or estimated (E) for the rivers in South (S.), Southeast (S.E.), and East (E.) Asia in comparison with those for the global rivers including all examined Asian rivers.

| River system | Mean (range) of $p$CO$_2$ (µatm) | | | | | Method | Reference |
|---|---|---|---|---|---|---|---|
| | Basin-wide | Mainstem | Headwater | Tributary | Impoundment | | |
| **Global** | 3100 (0-100000) | | | | | E | Raymond et al., 2013, GLORCIH[a] |
| | 2400 (2019–2826) | | | | | E | Lauerwald et al., 2015, GLORICH[a] |
| | | | 1116[b] (0-97906) | | | E | Marx et al., 2017, GLORICH[a] |
| **Asia (total)** | 1738 (3-11827) | 1217 (6-10977) | 819 (7-4646) | 2128 (3-11827) | 1089 (128-8785) | | This study (sum of only freshwater data below) |
| **S. Asia** | | | | | | | |
| Ganges | 891 (16-2778) | 1083 (26-2778) | 401 (92-1222) | 1597 (16-2778) | 181[c] | E/M | Manaka et al., 2015, GLORICH[a] |
| Brahmaputra | 605 (3-6706) | 396 (6-6706) | 29 (21-51) | 725 (3-3678) | | E | Huang et al., 2011, Manaka et al., 2015, Qu et al., 2017, GLORICH[a] |
| Indus | 694 (134-5907) | 660 (134-5907) | | 732 (165-3161) | | E | GLORICH[a] |
| Krishna | 2152 (180-4870) | 1871 (772-3631) | | 2305 (180-4870) | | E | GLORICH[a] |
| Godavari | 8785[d] (3944-16042) | | | | 8785[d] (3944-16042) | E | Prasad et al., 2013 |
| Bhote Kosi | 592 (35-5907) | | | | | E | GLORICH[a] |
| Various | 2685 (420-10977) | 3081 (426-10977) | | | 609 (420-692) | E/M | Panneer Selvam et al., 2014 |



| | | | | | | |
|---|---|---|---|---|---|---|
| Cochin estuary | (2975-6001)[e] | | | | E | Gupta et al., 2009 (only saline estuary data) |
| Other estuaries | 5882 (293-18492) | | | | E | Sarma et al., 2012 (only saline estuary data) |
| **S.E. Asia** | | | | | | |
| Mekong | 1235 (110-5970) | 1120 (224-5970) | 1367[f] | 1310 (110-4503) | 882[g] (864-899) | E/M | Alin et al., 2011, Li et al., 2013, Manaka et al., 2015b |
| Other rivers | 3586 (925-8555) | | | | | | Manaka et al., 2015b, Wit et al., 2015 |
| **E. Asia** | | | | | | |
| Yellow | 2257 (147-9659) | 2104 (423-4770) | 1082 (147-3546) | 2470 (511-9659) | 555[h], 450[i] (268-616) | E/M | Ran et al., 2015a, Ran et al., 2017a |
| Yangtze | 2341 (7-7718) | 1275 (528-2405) | 330 (7-3465) | 2675 (249-7718) | 1218 (693-1908) | E | Qu et al., 2015, Liu et al., 2016, Ran et al., 2017b[j], GLORICH[a] |
| Pearl River | 1679 (106-11000) | 2465 (600-7200) | 1527 (231-4646) | 1967 (106-11000) | | | Yao et al., 2007, Zhang et al., 2009, GLORICH[a] |
| Han River | 2783 (78-11827) | 1893 (78-8610) | 628 | 4560 (100-11827) | 251 (128-454) | M | Yoon et al., 2017 |

[a]GLORICH (Global River Chemistry Database; Hartmann et al., 2014), from which Raymond et al. (2013) and Lauerwald et al. (2015) calculated $pCO_2$ for global rivers, but excluded $pCO_2$ values with pH <5.4 and data with pH <5.4 and high pollution levels, respectively. Marx et al. (2017) excluded $pCO_2$ values above 100,000 μatm to avoid any potential overestimation resulting from pH and alkalinity effects; [b]The mean was calculated from the global mean $pCO_2$ (3100 μatm; Raymond et al., 2013) by assuming that headwaters account for 36% of the global riverine $CO_2$ emission (Marx et al., 2017); [c]Ganga Barrage at Rishikesh

5 (measured using headspace equilibration method; Park, unpublished data); [d]Dowleiswaram Reservoir (only the mean value was considered in summing data for Asia); [e]Cochin estuary: two sites on the Periyar River and 11 sites on the estuary; [f]Lancang headwater at Qinghai, China (measured using headspace equilibration method; Park, unpublished data); [g]Impounded tributaries of the Lower Mekong (Li et al., 2013); [h]Impoundment on the mainstem Yellow River (Ran et al., 2015a); [i]Impoundment on a Yellow River tributary (Ran et al., 2017a),; [j]Ran et al. (2017b) excluded $pCO_2$ values with pH <6.5.



**Figure 1**. Major river systems of South, Southeast, and East Asia that belong to top 30 global rivers based on discharge. The base map and the inset world map were modified from ArcGIS online Ocean Basemap and Milliman and Farnsworth (2011), respectively. Rivers addressed in the review and other large rivers are distinguished by different colors. Three Asian regions comprise the majority of Asian countries included in the regional categories "Asia" (indicated by a yellow color on the inset world map) and "Oceania" (dark green) used by Milliman and Farnsworth (2011).





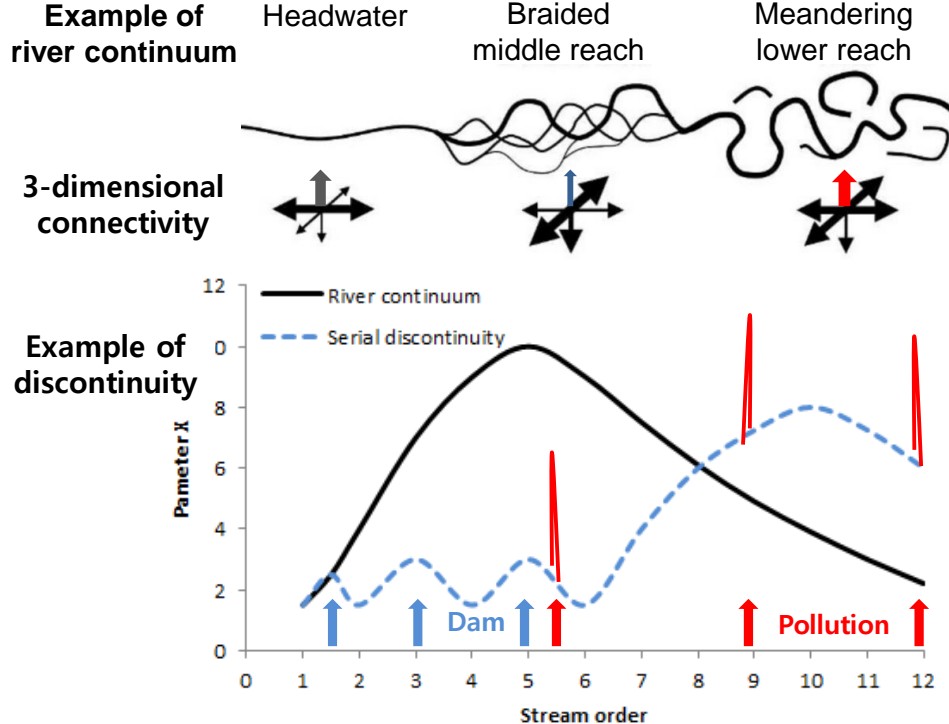

**Figure 2.** A schematic diagram describing river discontinuity in the anthropogenic land-water-scape: an example of river continuum observed in a minimally impacted river (top); three-dimensional connectivity vectors along the up-, mid-, and lower river reaches (middle); longitudinal variations in a hypothetical riverine biogeochemical process X (bottom). The connectivity vectors and the plots depicting river continuum and serial discontinuity were modified from Stanford and Ward (2001) and Poole (2010), complemented with some additional considerations including pollution-induced pulsatile discontinuities (red-colored pulses) and the vertical vector of air-water gas exchange (blue and red color indicating potential effects of dams and water pollution, respectively).



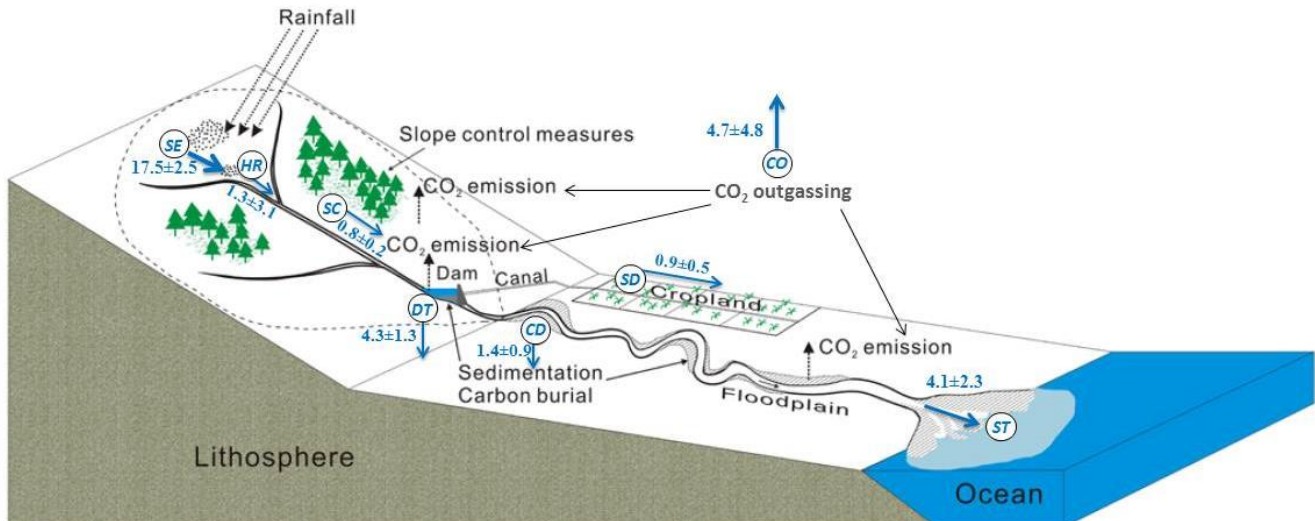

**Figure 3**. A schematic diagram illustrating human-induced alterations of the riverine C fluxes in the Yellow River as a model river system (modified from Ran et al., 2014). The annual flux rate (Tg C yr$^{-1}$) for each of the described soil and fluvial processes was estimated for the period 1950 – 2010. Slope and fluvial processes depicted in the figure include – SE: soil erosion, HR: hillslope redistribution, SC: slope control of erosion, DT: dam trapping, SD: sediment diversion, CD: channel deposition, CO: $CO_2$ outgassing, and ST: seaward transport of C. Refer to Ran et al. (2014) for more details on their flux and uncertainty estimations.





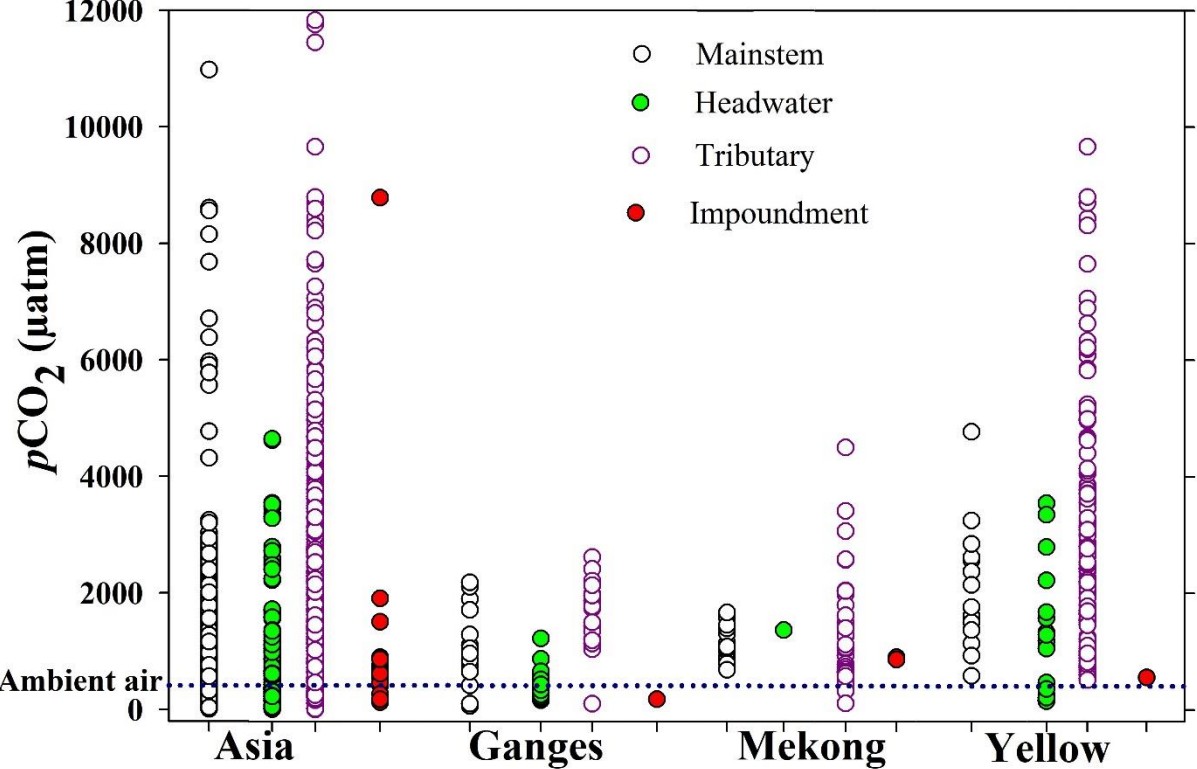

**Figure 4**. Comparison of available data of $pCO_2$ measured or estimated for the southern and eastern Asian rivers including three focal rivers of this study – the Ganges, the Mekong, and the Yellow River. The dotted line indicates the $pCO_2$ level of ambient air around 400 µatm. The number of data points included in the total Asian rivers: 1226 (mainstem: 278, headwater: 156, tributary: 773, impoundment: 19); the Ganges: 63 (mainstem: 14, headwater: 30, tributary: 18, impoundment: 1); the Mekong: 59 (mainstem: 19, headwater: 1, tributary: 37, impounded tributary: 2), the Yellow River: 201 (mainstem: 14, headwater: 22, tributary: 164, impoundment: 1). Refer to Table 3 for more details on data sources.




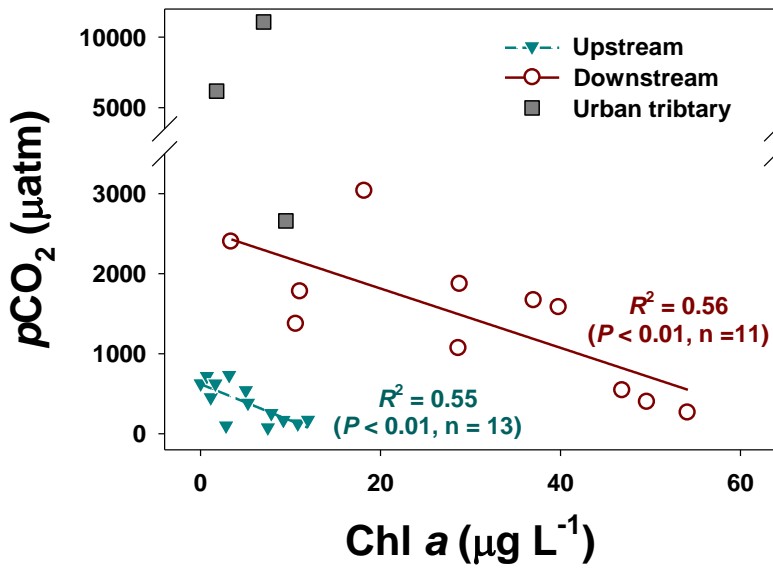

**Figure 5**. Longitudinal shift in the relationship between Chl *a* and $p$CO$_2$ observed between the up- and downstream reaches of the Han River receiving highly polluted urban tributaries (modified from the supplementary information of Yoon et al., 2017).