# Peer review of "Reviews and syntheses: Anthropogenic perturbations to carbon fluxes in Asian river systems: Concepts, emerging trends, and research challenges"

_Biogeosciences, 2017_

## Referee Comment (RC1) · Anonymous Referee #1 · 26 Feb 2018

This review article presents a very nice, pleasureable to read, comparative summary of CO2 fluxes in some of the largest Asian Rivers. Or, to be more correct, it presents a summary of the existing knowledge and of the challenges remaining as at present there is much less information on carbon fluxes in these large systems that what is available for the large river systems in the US or Europe. The review covers a good range of the published research in the chosen river systems (Mekong, Yellow and Ganges rivers) and also discusses some of the data from other systems in the region.

This leads me to my main comment: I was wondering why the authors have not discussed the Red River in Vietnam or Pearl River in China? There is a body of work on

both of these rivers. The authors don't need to go into great detail on the Red River, but I do think it merits a mention in the review.

I found the figures to be appropriate but perhaps the quality of Fig. 3. could be improved.

---

## Referee Comment (RC2) · Anonymous Referee #2 · 2 Mar 2018

Thank you for the opportunity to review this paper. Overall the authors provide a great deal of information and a nice compilation of existing information on Asian Rivers. The pace of development and ecosystem change is well emphasized and this is clearly an important topic and region of the world to focus on. The structure of the paper is quite weak, however, and the explanation of key concepts (especially pertaining to figure 2) must be improved to make the paper coherent. I hope my comments will help in this regard. My review is broken up into general and specific suggestions below.

General comments:

1. Throughout section 3 beginning on l. 149, this information is not clearly linked to

issues specific to Asian Rivers. I challenge the authors to improve this section and link it more strongly to outstanding issues in Asia.

2. The organization of the paper needs improvement. The manuscript could be strengthened if it were more tightly organized, shortened, and if there was less repetition. In sections 4 and 5 the authors break information up by flux pathways (Fig. 3 is a nice summary and example), but then repeatedly present findings for each region/river system and revisit previously discussed flux pathways. The discussion of one river system after another, without clear conceptual progression from one paragraph to the next, made it difficult for me to extract any generalizable information from these sections. The authors should do a better job grouping information. I think the obvious structure is grouping results from individual studies by pathways/fluxes, and explicitly defining how damming and pollution each appear to impact these fluxes (link back to figure 2).

3. The authors introduce their conceptual figure 2 then make little effort to integrate it in sections 4 and 5. This must really be brought in, both to guide readers conceptually, but also to provide evidence that what they are suggesting is happening in figure 2 is actually supported with existing data.

4. There is no clear summary of section 4. The section is quite long, with much anecdotal information. I encourage the authors to end by brining this information together to make a more clear statement as to what current information suggests impoundment is actually doing to C fluxes in Asian Rivers. Brining the discussion back to their conceptual figure 2 would especially strengthen the paper.

5. The authors repeatedly lead paragraphs by pointing out that Asian systems are undersampled. I think this paper would be more useful to readers if the authors instead presented a summary of the state of knowledge from existing literature, then explained what is missing or has large associated uncertainty, then either presented a new synthesis of information or concluded with a statement of what needs to be done in the

future. For example, on line 126 the authors criticize the outdated existing C budgets, but do not follow up with any new information or an updated budget. Another example is found on line 297/8, where the authors say that few systematic studies exist, then proceed to cite 2 such efforts. To me it makes more sense to restructure these paragraphs to lead in with what is known, then either refine these estimates, or lay out in a clear way what is needed to refine this understanding.

6. Throughout the paper, the authors finish many paragraphs without a clear concluding statement to summarize the pgph. I point out a number of instances below, but the authors really need to go through the paper and try to fix this.

7. Summary section: The title of the paper includes 'concepts and emerging trends". Neither of these themes are really revisited in the summary. Instead, new topics are introduced here that should not be (i.e., monsoons and climate are introduced, then more site specific trends are presented, and CO2 sampling methodology is brought up). This section really needs to link back to figure 2, it needs to summarize what was discussed in sections 4 and 5, and needs to lay out existing gaps in our understanding of C cycling in Asian Rivers (more than just saying we need more sampling and coverage).

8. I get the sense that this is a pCO2-focused paper, but CH4 data are thrown in here at random. I would consider removing any discussion of CH4.

Specific comments:

L21. Objective, not object

L22. Change 'a latest update' to 'an update'

L30. Change to 'vary greatly'

L32. Change to 'The rapid'

L52. This pgph contains multiple themes (Global C cycle, Asian Rivers, waste water

effluent). Please restructure to 1 theme per paragraph.

L85. Remove 'the' from 'the riverine'

L105. This pgph is off topic and should not be in section 2. Consider integrating this information with the introduction.

L111. Remove 'either'

L125. The pgph lacks a concluding sentence.

L135-138. Remove this information, it is off topic.

L146. How does this pgph relate to the theme of section 2? Add a concluding sentence that links it back to the question.

L160. Be specific. How are hydroclimates and aquatic ecosystems affected? Consider rephrasing.

L166. Latitudinal? Unclear

L198. Better introduce/explain serial discontinuity

L210. Be specific. Rather than quoting here, please explain what you mean.

L229-31. Not a great end here. You just said most changes are due to human activity so this is a little inconsistent. You need to cite actual data and numbers to quantitatively compare these effects.

L232. Improve sub-heading: Effect of river impoundments on what specifically?

L246. Affected how? Be specific.

L264-265. This information is contradictory. Which is it? A stable C sink, a site of intense OC conversion, or both simultaneously? please rephrase.

L275 and 276. Turbulence?

L279. Weak conclusion. Add a sentence to summarize pgph.

L320. Weak conclusion. Add a sentence to summarize pgph.

L321. This pgph is just a series of anecdotes. What is the point?

L336. Again, effects on what? The C cycle? pCO2? Be specific.

L335. More of an introduction pgph. Consider moving this section should be carbon-specific.

L368. Conclusion sentence? Are you saying that labile OM inputs appear to boost CO2 pool?

L385. I'd cite fig. 3 instead of fig 4 here.

L401. Can you link this back to figure 4 and actual CO2 trends?

L495. Be specific. What kind of alterations?

L503. Be specific. What contrasts?

Figure 4: Consider presenting boxplots, this would help readers summarize the data. Figure 5: Completely underused. This figure needs to be cited more throughout the text.

---

## Author Comment (AC1) · 30 Mar 2018

Please also see our author response in the attached appendix file including the same text and a revised figure.

Author response to RC1

This review article presents a very nice, pleasureable to read, comparative summary of CO2 fluxes in some of the largest Asian Rivers. Or, to be more correct, it presents a summary of the existing knowledge and of the challenges remaining as at present there is much less information on carbon fluxes in these large systems that what is available

for the large river systems in the US or Europe. The review covers a good range of the published research in the chosen river systems (Mekong, Yellow and Ganges rivers) and also discusses some of the data from other systems in the region. <Response> We thank the reviewer for the very positive evaluation of our manuscript.

This leads me to my main comment: I was wondering why the authors have not discussed the Red River in Vietnam or Pearl River in China? There is a body of work on both of these rivers. The authors don't need to go into great detail on the Red River, but I do think it merits a mention in the review. <Response> We have cited more papers on the Red and Pearl Rivers in relevant sections, including our data synthesis (Table 3; Fig. 4).

I found the figures to be appropriate but perhaps the quality of Fig. 3. could be improved. <Response> We have improved Fig. 3 by changing the font type and size of labels and the color of some figure components (see the revised figure in the appendix file).

Please also note the supplement to this comment:
https://www.biogeosciences-discuss.net/bg-2017-549/bg-2017-549-AC1-supplement.pdf

---

## Author Comment (AC2) · 30 Mar 2018

"Please also see our response in the appendix file that includes the same text, a revised figure, and the revised manuscript."

Author response to RC2

Thank you for the opportunity to review this paper. Overall the authors provide a great deal of information and a nice compilation of existing information on Asian Rivers. The pace of development and ecosystem change is well emphasized and this is clearly an important topic and region of the world to focus on. The structure of the paper

is quite weak, however, and the explanation of key concepts (especially pertaining to figure 2) must be improved to make the paper coherent. I hope my comments will help in this regard. My review is broken up into general and specific suggestions below. <Response> We thank the reviewer for the very detailed and really insightful comments, which have helped us a lot to improve the coherence and readability of the manuscript. As addressing the reviewer comments requires a substantial rewriting of the entire manuscript, we had to revise the manuscript first to respond to the comments more efficiently, though we are aware that at this stage we need to provide only our revision plans. The revision of the sentences and paragraphs pointed by the reviewer was indicated in the enclosed manuscript by a dark blue color, whereas additional revisions indirectly associated with the reviewer comments were marked in a light blue color.

General comments: 1. Throughout section 3 beginning on l. 149, this information is not clearly linked to issues specific to Asian Rivers. I challenge the authors to improve this section and link it more strongly to outstanding issues in Asia. <Response> This section was conceived as a general conceptual framework on which later sections build to address specific issues relevant for Asian river systems. To use the proposed framework more coherently and "visibly" as a common thread connecting section 3 and the following sections, we added more sentences indicating the relevance and implications of the addressed concepts for Asian rivers. Please refer to the revised section. One example is provided below. Lines 166-168: Rapid, concurrent changes in land use and river flow and chemistry make Asian rivers a perfect test bed for exploring how human-induced perturbations alter hydro-biogeochemical cycles across the components of these anthropogenic land-water-scapes.

2. The organization of the paper needs improvement. The manuscript could be strengthened if it were more tightly organized, shortened, and if there was less repetition. In sections 4 and 5 the authors break information up by flux pathways (Fig. 3 is a nice summary and example), but then repeatedly present findings for each region/river system and revisit previously discussed flux pathways. The discussion of one river system after another, without clear conceptual progression from one paragraph to the next, made it difficult for me to extract any generalizable information from these sections. The authors should do a better job grouping information. I think the obvious structure is grouping results from individual studies by pathways/fluxes, and explicitly defining how damming and pollution each appear to impact these fluxes (link back to figure 2). <Response> The entire manuscript was thoroughly revised to remove redundant descriptions (e.g., repeated descriptions on the lack of data and less important references) and reorganize information based on the key message (e.g., two background paragraphs in the introduction and many parts of section 4 and 5). With regard to the paragraphs on three river systems (the Ganges, Mekong, and Yellow River) selected as representative systems of three reviewed Asian regions (as now clearly stated in the abstract and introduction), descriptions of case studies were shortened and tuned to the main conceptual framework introduced in section 3 (discontinuities created by river impoundment and pollution). Please refer to the revised sections and understand that all the changes cannot be described here.

3. The authors introduce their conceptual figure 2 then make little effort to integrate it in sections 4 and 5. This must really be brought in, both to guide readers conceptually, but also to provide evidence that what they are suggesting is happening in figure 2 is actually supported with existing data. <Response> To bring in the conceptual framework coherently throughout the manuscript, we explicitly stated the limitation of the conventional river continuum concept (L 190-193) and details of the addressed concepts (section 2), reorganized sections 4 and 5, rewrote the concluding paragraph in section 6 (L 534-549). Again, please refer to the revised sections and understand that all the changes cannot be described here.

4. There is no clear summary of section 4. The section is quite long, with much anecdotal information. I encourage the authors to end by bringing this information together to make a more clear statement as to what current information suggests impoundment is actually doing to C fluxes in Asian Rivers. Brining the discussion back to their conceptual figure 2 would especially strengthen the paper. <Response>The last paragraph, together with the paragraphs describing detailed cases for the three selected rivers, was modified as a summary of impoundment effects in line with the conceptual frame. L 338-356: "In accordance with the serial discontinuity concept (Ward and Standford, 1983; Fig. 2), multiple dams constructed on large Asian rivers such as the Mekong and Yellow River create standing water conditions that may shift stream metabolisms and $pCO_2$ dynamics from the patterns observed for freely flowing reaches...The paucity of $pCO_2$ measurements in dammed Asian rivers (Table 3; Fig. 4) does not allow for any generalization of long-term impoundment effects on sediment C storage and $CO_2$ emissions, demanding more long-term investigations of seasonal and year-to-year variations in metabolic processes and $pCO_2$ levels across a wide range of impounded inland water systems."

5. The authors repeatedly lead paragraphs by pointing out that Asian systems are undersampled. I think this paper would be more useful to readers if the authors instead presented a summary of the state of knowledge from existing literature, then explained what is missing or has large associated uncertainty, then either presented a new synthesis of information or concluded with a statement of what needs to be done in the future. For example, on line 126 the authors criticize the outdated existing C budgets, but do not follow up with any new information or an updated budget. Another example is found on line 297/8, where the authors say that few systematic studies exist, then proceed to cite 2 such efforts. To me it makes more sense to restructure these paragraphs to lead in with what is known, then either refine these estimates, or lay out in a clear way what is needed to refine this understanding. <Response> Throughout the manuscript, repeated descriptions of data paucity were minimized and redundant sentences were replaced by more specific descriptions of research needs. The paragraph describing outdated C budgets on L 126 has been reorganized so the second paragraph of section 2 (L 123-134) now summarizes old and new budgets and emerging regional trends and research needs are provided in the following paragraph

(L 135-150).

6. Throughout the paper, the authors finish many paragraphs without a clear concluding statement to summarize the pgph. I point out a number of instances below, but the authors really need to go through the paper and try to fix this. <Response> The manuscript has thoroughly been revised to fix any incomplete paragraphs including those pointed by the reviewer. Please see our detailed responses below.

7. Summary section: The title of the paper includes 'concepts and emerging trends". Neither of these themes are really revisited in the summary. Instead, new topics are introduced here that should not be (i.e., monsoons and climate are introduced, then more site specific trends are presented, and CO2 sampling methodology is brought up). This section really needs to link back to figure 2, it needs to summarize what was discussed in sections 4 and 5, and needs to lay out existing gaps in our understanding of C cycling in Asian Rivers (more than just saying we need more sampling and coverage). <Response> The section has been reorganized to summarize emerging trends on two major topics (L 495-514: impoundment effects; L 515-533: urbanization/pollution effects) and provide concluding remarks on the observed trends in line with the conceptual framework in the last paragraph (L 534-549)

8. I get the sense that this is a pCO2-focused paper, but CH4 data are thrown in here at random. I would consider removing any discussion of CH4. <Response> Any irrelevant descriptions on CH4 have been removed from the revised manuscript.

Specific comments: L21. Objective, not object <Response> A shortened sentence is now used (L 21: "This review aims to provide...")

L22. Change 'a latest update' to 'an update' <Response> Changed. L30. Change to 'vary greatly' <Response> Changed.

L32. Change to 'The rapid' <Response> Changed.

L52. This pgph contains multiple themes (Global C cycle, Asian Rivers, waste water

effluent). Please restructure to 1 theme per paragraph. <Response> The first and following two paragraphs have been restructured into two paragraphs: the first one (L 56-67) on the lack of Asian river data that limits our ability to estimate the global C budget, and the second one (L 68-84) on anthropogenic perturbations including water pollution and dam issues focusing on Asian rivers.

L85. Remove 'the' from 'the riverine' <Response> Removed.

L105. This pgph is off topic and should not be in section 2. Consider integrating this information with the introduction. <Response> This paragraph was added during the technical review in response to the editor's suggestion for specifying the geographical extent of Asian rivers. To keep the revised introduction compact and make section 2 more coherent, the section title has been changed to "The geographical scope, global implications, and emerging regional trends of Asian river systems".

L111. Remove 'either' <Response> Removed.

L125. The pgph lacks a concluding sentence. <Response> The paragraph (L 122-134) has been restructured and added with a concluding remark (see below) emphasizing the importance of a regional synthesis of DOC, POC, and DIC for evaluating Asian monsoon rivers to the global riverine C fluxes. L 130-134: "Monsoonal increases in discharge and POC can have either a positive effect on riverine pCO2 levels through enhanced soil flushing of DIC and/or in-stream organic C biodegradation or a negative effect caused by dilution, as observed in such turbid Asian rivers as the Pearl (Yao et al., 2007), Yangtze (Li et al., 2012), and Mekong (Li et al., 2013). Constraining differential monsoon effects on the fluxes of DOC, POC, and DIC including CO2 represents a key challenge in evaluating the contribution of Asian rivers to the global riverine C fluxes."

L135-138. Remove this information, it is off topic. <Response> The sentence has been removed.

L146. How does this pgph relate to the theme of section 2? Add a concluding sentence that links it back to the question. <Response> This paragraph and the one before will be combined into one paragraph explaining major environmental changes affecting Asian rivers. A concluding sentence has been added to mention that this review focuses on river impoundment and eutrophication will not address land use change issue in detail. L 146-150: Deforestation and associated peatland drainage in tropical areas represent another important, but rarely explored topic with regard to $CO_2$ outgassing from Asian rivers (Baum et al., 2007; Wit et al., 2015). A recent study suggested that peatland drainage could enhance organic matter degradation in the coastal peatlands and organic rich soils of Southeast Asian lowland areas and islands, increasing $CO_2$ outgassing from the rivers draining the affected areas (Wit et al., 2015). However, this issue cannot be addressed in detail here, because only a few studies have been conducted in Indonesia and Malaysia.

L160. Be specific. How are hydroclimates and aquatic ecosystems affected? Consider rephrasing. <Response> The sentence has been split and the second sentence explains effects on flow and downstream ecosystems specifically. L 164-166: In the case of rivers draining arid areas, WWTP effluents can not only increase river flow but also provide a source of water feeding into habitats for various aquatic organisms along downstream reaches.

L166. Latitudinal? Unclear <Response> The sentence has been rephrased. L 172-174: The original river continuum concept envisaged gradual and continual changes in OM composition and metabolic rates in correspondence to downstream variations in environmental conditions and biotic communities along the river (Vannote et al., 1980; Fig. 2).

L198. Better introduce/explain serial discontinuity <Response> Serial discontinuity is defined in a previous sentence (L 199-200: which states that stream regulation by multiple dams results in "an alternating series of lentic and lotic reaches") and this sentence has been rephrased to describe the limitation of the concept, particularly for polluted Asian rivers (L 205-208: Although serial discontinuity concept has been a use-

ful framework for assessing anthropogenic impacts on regulated lotic systems, its pre-suppositions including no disturbances other than impoundment (Ward and Stanford, 1983) limit its application to investigating other environmental stresses than impoundments, such as high levels of organic pollutants and nutrients observed in many Asian rivers receiving untreated sewage and urban runoff.).

L210. Be specific. Rather than quoting here, please explain what you mean. <Response> The sentence has been rewritten L 217-219: As Hynes (1975) emphasized the importance of the terrestrial-aquatic connectivity in headwater systems by saying that "the valley rules the stream, human-induced changes in the watershed would have large cascading effects on the structure and function of stream ecosystems.

L229-31. Not a great end here. You just said most changes are due to human activity so this is a little inconsistent. You need to cite actual data and numbers to quantitatively compare these effects. <Response> New concluding sentences have been added to emphasize the importance of the interplay between human-induced perturbations and the concurrent climate change. L 239-245: In many dammed Asian rivers, observed decreases in discharge and sediment transport might be largely explained by increasing river impoundments and water diversion (Milliman et al., 2008; Li and Bush et al., 2016). However, recent increases in the frequency and intensity of extreme precipitation events observed in many parts of Asia (Min et al., 2011) suggest that potential changes in monsoon rainfall regimes as a consequence of climate change can amplify seasonal and year-to-year variations in discharge and the transport of sediment and C even in dammed river systems. Therefore, predicting future changes of riverine C fluxes in increasingly human-modified Asian river systems would require a better understanding of the complex interplay between anthropogenic perturbations and the concurrent climate change.

L232. Improve sub-heading: Effect of river impoundments on what specifically? <Response> The sub-heading has been changed to " Contrasting effects of river impoundment on organic C transport and CO2 emission" (L 247).

L246. Affected how? Be specific. <Response> The sentence has been reformulated and followed by a new, concluding sentence to put the Asian dam issue in the global context. L 260-262: Many Asian rivers, such as the Indus, Yangtze, and Yellow, have seen largest reductions in sediment export to the oceans compared to the pre-dam era (Syvitski et al., 2005). Therefore, investigating altered rates of C storage and losses in dammed Asian rivers is crucial for a better understanding of human impacts on global riverine C fluxes.

L264-265. This information is contradictory. Which is it? A stable C sink, a site of intense OC conversion, or both simultaneously? please rephrase. <Response> The sentence has been rephrased (see below) and followed by the contrasting findings on the source or sink function of sediments in the Yellow River basin. L 279-280: It remains an important research question whether POC trapped in ever-growing reservoir sediments within the Yellow River basin would function as a sink or source of $CO_2$ for the atmosphere (Zhang et al., 2013; Ran et al., 2015a, 2017a).

L275 and 276. Turbulence? <Response> To make clear the comparison between standing reservoir waters and flowing waters up- and downstream, the sentence has been rephrased. L 290-293: Compared to the standing waters of the reservoirs with enhanced primary production, rapidly flowing waters along both the upstream and downstream reaches were larger C sources for the atmosphere, exhibiting much higher $pCO_2$ levels and faster flow velocities providing favorable conditions for an efficient gas evasion from the aqueous boundary layer.

L279. Weak conclusion. Add a sentence to summarize pgph. <Response> A concluding sentence has been added. L 296-298: The contrasting impoundment effects observed in the Yellow River and other Chinese rivers suggest that a basin-wide assessment of impoundment impacts on sediment C storage and $CO_2$ emissions should take into consideration concurrent changes in primary production and organic matter biodegradation.

L320. Weak conclusion. Add a sentence to summarize pgph. <Response> A concluding sentence has been added, following a thorough revision of the entire paragraph on the Indian river case (L 326-334). L 334-337: It demands further research to establish how seasonal and inter-annual variations in climatic and trophic conditions in dammed Indian rivers alter the balance between autotrophy and heterotrophy and hence CO2 emissions along the "discontinuous" river-reservoir-estuary continuum as found in the Godavari basin.

L321. This pgph is just a series of anecdotes. What is the point? <Response> To make a clear point (impoundment shifts metabolisms and CO2 dynamics from flowing water conditions) in this summary paragraph, the serial discontinuity concept was brought in the beginning sentence and the contrasting impoundment effects observed in the reviewed rivers were explained by the changing balance between competing processes over time (L 338-344). The paragraph has also been added with concluding remarks (L 353-356). L 338-344: In accordance with the serial discontinuity concept (Ward and Standford, 1983; Fig. 2), multiple dams constructed on large Asian rivers such as the Mekong and Yellow River create standing water conditions that may shift stream metabolisms and pCO2 dynamics from the patterns observed for freely flowing reaches. The observed contrasting impoundment effects on CO2 emission across different Asian river systems might have resulted from an interplay between planktonic CO2 uptake, organic matter biodegradation, and sediment C sequestration (Liu et al., 2016; Maavara et al., 2017). The balance between the competing processes affecting the actual level of pCO2 in reservoir waters may change not only seasonally (Prasad et al., 2013; Liu et al., 2016) but also with the increasing age of dams (Barros et al., 2011). L 353-356: The paucity of pCO2 measurements in dammed Asian rivers (Table 3; Fig. 4) does not allow for any generalization of long-term impoundment effects on sediment C storage and CO2 emissions, demanding more long-term investigations of seasonal and year-to-year variations in metabolic processes and pCO2 levels across a wide range of impounded inland water systems. . L336. Again, effects on what? The C cycle? pCO2? Be specific. <Response> The sub-heading has been changed

to "Effects of water pollution on riverine metabolisms and CO2 emissions".

L335. More of an introduction pgph. Consider moving this section should be carbon-specific. <Response> The first sentence has been moved to Introduction (L 72-74) and replaced by a short beginning sentence (L 359: Across Asia rapidly urbanizing river basins are highly polluted with poorly treated or untreated wastewater.) We had to keep the remaining part here not to have a too detailed introduction. We also wanted to provide a background on the current status of water pollution in Asian rivers before addressing main carbon-specific issues in the following paragraphs.

L368. Conclusion sentence? Are you saying that labile OM inputs appear to boost CO2 pool? <Response> The sentence has been rephrased as a concluding remark on labile OM of anthropogenic origin boosting microbial processing of riverine OM and CO2 emission. L 470-473: These chemical analyses, together with direct underway measurements that revealed extraordinarily high pCO2 levels along polluted river and estuarine reaches of some Chinese rivers (Zhai et al., 2005; Chou et al., 2013; Wang et al., 2017), suggest that labile OM fractions of anthropogenic origin can boost microbial processing of the bulk riverine OM, enhancing CO2 emissions from polluted waterways.

L385. I'd cite fig. 3 instead of fig 4 here. <Response> Fig. 3 has been cited instead of Fig. 4 (L 387).

L401. Can you link this back to figure 4 and actual CO2 trends? <Response> The entire paragraph has been rewritten, now citing Fig. 4 and ending up with some concluding remarks. L 394-433: As a combined result of higher loads of pollutants discharged from local sources to tributaries and higher flow diluting pollutant concentrations in the mainstem, tributaries appear to be more polluted than the mainstem Yellow River, exhibiting the highest levels of pCO2 among the three compared river systems (Fig. 4). Higher concentrations of DOC and POC in more polluted lower reaches and their tributaries might lead to enhanced in-stream biodegradation of allochthonous C by labile OM fractions of anthropogenic origin, but altered rates of biodegradation and primary

production have not yet been measured in any reach of the Yellow River. Along with the question about impoundment effects on sediment C, the role of organic pollutions in riverine metabolisms and CO2 emissions along lower reaches is crucial for understanding the fate of organic C derived from various sources in the Yellow River basin including C stocks stored in reservoir and floodplain sediments.

L495. Be specific. What kind of alterations? <Response> The sentence has been rephrased (see below). More details on specific alterations are provided in the revised paragraphs. L 495-496: This review identified alarming regional trends concerning dam construction booms and the rapid pace of urbanization across three reviewed Asian regions, both of which can significantly alter riverine metabolisms and C dynamics.

L503. Be specific. What contrasts? <Response> The sentence has been rephrased. L 505-508: Unlike in Europe and North America where very few large dam projects have been commissioned over the recent decades, the current booms of mega dam construction across Asia appear to induce ever-increasing perturbations to riverine C fluxes, demanding more systematic assessments of impoundment impacts on riverine organic C transport and GHG emissions.

Figure 4: Consider presenting boxplots, this would help readers summarize the data. <Response> Figure 4 has been replaced by a new version presenting boxplots (see the revised version in the attached appendix file).

Please also note the supplement to this comment:
https://www.biogeosciences-discuss.net/bg-2017-549/bg-2017-549-AC2-supplement.pdf
* * *